# Rayleigh Quotient Graph Neural Networks for Graph-level Anomaly Detection

**Xiangyu Dong, Xingyi Zhang, Sibo Wang**
Department of Systems Engineering and Engineering Management
The Chinese University of Hong Kong
{xydong, xyzhang, swang}@se.cuhk.edu.hk

## Abstract

Graph-level anomaly detection has gained significant attention as it finds applications in various domains, such as cancer diagnosis and enzyme prediction. However, existing methods fail to capture the spectral properties of graph anomalies, resulting in unexplainable framework design and unsatisfying performance. In this paper, we re-investigate the spectral differences between anomalous and normal graphs. Our main observation shows a significant disparity in the accumulated spectral energy between these two classes. Moreover, we prove that the accumulated spectral energy of the graph signal can be represented by its Rayleigh Quotient, indicating that the Rayleigh Quotient is a driving factor behind the anomalous properties of graphs. Motivated by this, we propose *Rayleigh Quotient Graph Neural Network (RQGNN)*, the first spectral GNN that explores the inherent spectral features of anomalous graphs for graph-level anomaly detection. Specifically, we introduce a novel framework with two components: the Rayleigh Quotient learning component (RQL) and Chebyshev Wavelet GNN with RQ-pooling (CWGNN-RQ). RQL explicitly captures the Rayleigh Quotient of graphs and CWGNN-RQ implicitly explores the spectral space of graphs. Extensive experiments on 10 real-world datasets show that RQGNN outperforms the best rival by 6.74% in Macro-F1 score and 1.44% in AUC, demonstrating the effectiveness of our framework. Our code is available at https://github.com/xydong127/RQGNN.

## 1 Introduction

Graph-structure data explicitly expresses complex relations between items, and thus has attracted much attention from the deep learning community. Extensive efforts have been devoted to deploying GNNs (Kipf & Welling, 2017; Hamilton et al., 2017; Velickovic et al., 2018) on node-level tasks. Recently, researchers have started to shift their focus from local properties to graph-level tasks (Wang et al., 2021; Liu et al., 2022; Yue et al., 2022), and graph-level anomaly detection has become one of the most important graph-level tasks with diverse applications (Ma et al., 2022; Zhang et al., 2022; Qiu et al., 2022), such as cancer diagnosis, enzyme prediction, and brain disease detection. In addition, applications of graph-level anomaly detection can be observed in trending topics, such as spam detection (Li et al., 2019) and rumor detection (Bian et al., 2020).

Following the common design of graph learning models, existing solutions for graph-level anomaly detection mainly employ spatial GNNs with distinct pooling techniques. For example, CAL (Sui et al., 2022) and FAITH (Wang et al., 2022) incorporate node features with topological characteristics of graphs to generate graph representations. Meanwhile, due to the limitations of the average or sum pooling function in certain tasks, researchers have introduced various graph pooling functions (Wu et al., 2022; Hua et al., 2022; Liu et al., 2023). However, to the best of our knowledge, no previous attempt has provided spectral analysis for anomalous graphs, missing an important feature that can help better capture the properties of anomalous graphs.

To address this issue, we start by investigating the spectral energy of the graph Laplacian. Our key findings and theoretical analysis validate that the accumulated spectral energy can be represented by the Rayleigh Quotient, which has been studied in the physics and mathematics area (Pierre, 1988; Chan et al., 2011). Besides, Tang et al. (2022) makes an interesting observation related to the

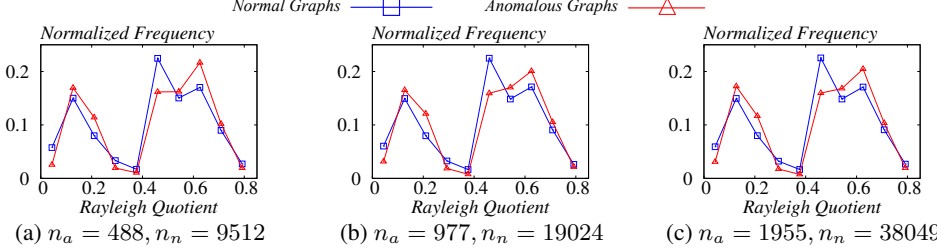

Figure 1: Normalized Rayleigh Quotient distribution on SN12C.

Rayleigh Quotient for node anomaly detection. However, the inherent properties of the Rayleigh Quotient in the graph domain are still relatively under-explored. Hence, we investigate the Rayleigh Quotient for graphs and further empirically show that the Rayleigh Quotient distributions of normal graphs and anomalous graphs follow different patterns. In particular, we first randomly sample $n_a$ anomalous graphs and $n_n$ normal graphs. For each graph, we calculate its corresponding Rayleigh Quotient. Subsequently, we set the maximum and minimum values of the Rayleigh Quotient of graphs as the bounds of the value range, which is then divided into 10 equal-width bins. After that, we assign each value of the Rayleigh Quotient of graphs to its corresponding bin. Finally, we calculate the frequency of values that fall into each bin and normalize them, which can be regarded as the normalized Rayleigh Quotient distribution of the sampled dataset. Figure 1 reports the Rayleigh Quotient distribution on the SN12C dataset, and the results on other datasets can be found in Appendix A.9. As we can observe, regardless of the variations in the sample size, the Rayleigh Quotient distribution of each class exhibits a consistent pattern across different sample sets. In addition, it is evident from Figure 1 that the Rayleigh Quotient distribution of anomalous graphs and that of normal ones are distinct from each other statistically. This observation highlights how the Rayleigh Quotient can reveal the underlying differences between normal and anomalous graphs. Hence, the Rayleigh Quotient should be encoded and explored when identifying anomalous graphs. Additionally, as we establish a connection between the Rayleigh Quotient and the spectral energy of the graph Laplacian, it becomes apparent that the spectral energy distribution exhibits robust statistical patterns. This, in turn, empowers us to leverage spectral graph neural networks for further encoding and utilization of this valuable information.

Motivated by the observation and theoretical analysis, in this paper, we propose RQGNN, a Rayleigh Quotient-based GNN framework for graph-level anomaly detection tasks. It consists of two main components: the Rayleigh Quotient learning component (RQL) and Chebyshev Wavelet GNN with RQ-pooling (CWGNN-RQ). Firstly, we adopt RQL to derive the Rayleigh Quotient of each graph and then employ a multi-layer perceptron (MLP) to generate the representation of each graph, aiming to capture explicit differences between anomalous and normal graphs guided by their Rayleigh Quotient. Secondly, to obtain the implicit information embedded in the spectral space, we draw inspiration from the Chebyshev Wavelet GNN (CWGNN) and adopt it to learn the inherent information in the graph data. Besides, to alleviate the drawbacks of existing pooling techniques in graph-level anomaly detection, we introduce a powerful spectral-related pooling function called RQ-pooling. Furthermore, we address the challenge of imbalanced data in graph-level anomaly detection via a class-balanced focal loss. The final graph embedding is the combination of representations generated by the RQL and CWGNN-RQ. By combining the explicit information from the Rayleigh Quotient and the implicit information from the CWGNN-RQ, RQGNN effectively captures more inherent information for the detection of anomalous graphs.

In our experiments, we evaluate RQGNN against 10 alternative frameworks across 10 datasets. Extensive experiments demonstrate that our proposed framework consistently outperforms spectral GNNs and the state-of-the-art (SOTA) GNNs for both graph classification task and graph-level anomaly detection task. We summarize our contributions as follows:

- Our main observation and theoretical analysis highlight that the Rayleigh Quotient reveals underlying properties of graph anomalies, providing valuable guidance for future work in this field.
- We propose the first spectral GNNs for the graph-level anomaly detection task, which incorporates explicit and implicit learning components, enhancing the capabilities of anomaly detection.
- Comprehensive experiments show that RQGNN outperforms SOTA models on 10 real-world graph datasets, demonstrating the effectiveness of RQGNN.

## 2 Preliminaries

**Notation.** Let $G = (\boldsymbol{A}, \boldsymbol{X})$ denote a connected undirected graph with $n$ nodes and $m$ edges, where $\boldsymbol{X} \in \mathbb{R}^{n \times F}$ is node features, and $\boldsymbol{A} \in \mathbb{R}^{n \times n}$ is the adjacency matrix. We set $\boldsymbol{A}_{ij} = 1$ if there exists an edge between node $i$ and $j$, otherwise $A_{ij} = 0$. Let $\boldsymbol{D}$ be the diagonal degree matrix, the Laplacian matrix $\boldsymbol{L}$ is then defined as $\boldsymbol{D} - \boldsymbol{A}$ (regular) or as $\boldsymbol{I_n} - \boldsymbol{D}^{-\frac{1}{2}} \boldsymbol{A} \boldsymbol{D}^{-\frac{1}{2}}$ (normalized), where $\boldsymbol{I}_n$ is an $n \times n$ identity matrix.

**Rayleigh Quotient.** The regular Laplacian matrix can be decomposed as $\boldsymbol{L} = \boldsymbol{U \Lambda U}^T$, where $\boldsymbol{U} = (\boldsymbol{u}_1, \boldsymbol{u}_2, ..., \boldsymbol{u}_n)$ represents orthonormal eigenvectors and the corresponding eigenvalues are sorted in ascending order, i.e. $\lambda_1 \leq ... \leq \lambda_n$. Let $\boldsymbol{x} = (x_1, x_2, ..., x_n)^T \in \mathbb{R}^n$ be a signal on graph $G$, $\hat{\boldsymbol{x}} = (\hat{x_1}, \hat{x_2}, ..., \hat{x_n})^T = \boldsymbol{U}^T \boldsymbol{x}$ is the graph Fourier transformation of $\boldsymbol{x}$. Following the definition in Horn & Johnson (2012), the Rayleigh Quotient is defined as $\frac{\boldsymbol{x}^T \boldsymbol{L} \boldsymbol{x}}{\boldsymbol{x}^T \boldsymbol{x}}$.

Next, we briefly review spectral GNNs and existing work for graph-level anomaly detection and graph classification.

**Spectral GNN.** By processing Laplacian matrix, spectral GNNs manipulate the projection of graph spectrum (Defferrard et al., 2016) and can be viewed as graph signal processing models. It has drawn much attention in the graph learning community. For instance, ChebyNet (Defferrard et al., 2016) and BernNet (He et al., 2021) utilize different approximations of spectral filters to improve the expressiveness of spectral GNN. Specformer (Bo et al., 2023) combines the transformer and spectral GNN to perform self-attention in the spectral domain. BWGNN (Tang et al., 2022) adopts a wavelet filter to generate advanced node representations for node-level anomaly detection, showing the potential ability of wavelet filter in the anomaly detection area.

**Graph-level Anomaly Detection.** To the best of our knowledge, OCGIN (Zhao & Akoglu, 2021) is the first to explore graph-level anomaly detection, which provides analysis to handle the performance flip of several methods on graph classification datasets. After that, OCGTL (Qiu et al., 2022) adopts graph transformation learning to identify anomalous graphs and GLocalKD (Ma et al., 2022) investigates the influence of knowledge distillation on graph-level anomaly detection. A following work, HimNet (Niu et al., 2023) builds a hierarchical memory framework to balance the anomaly-related local and global information. One recent study, iGAD (Zhang et al., 2022) suggests that the anomalous substructures lead to graph anomalies. It proposes an anomalous substructure-aware deep random walk kernel and a node-aware kernel to capture both topological and node features, achieving SOTA performance. Yet, existing solutions only explain the anomalous phenomena from spatial perspectives. In contrast, our RQGNN further explores the spectral aspects of anomalous graphs, leading to an explainable model design and satisfying model performance.

**Remark**. In out-of-distribution datasets, only one class of in-distribution data is given, while anomaly datasets usually contain data with two distinct characteristics. Therefore, out-of-distribution detection focuses on determining whether a new sample belongs to the given class, while anomaly detection aims to determine a new sample belongs to which class.

**Graph Classification.** Graph classification models can also be considered as a general framework for our task. GMT (Baek et al., 2021) points out that a simple sum or average pooling function is unlikely to fully collect information for graph classification. The authors propose a multi-head attention-based global pooling layer to capture the interactions between nodes and the topology of graphs. Afterward, Gmixup (Han et al., 2022) applies data augmentation to improve the generalization and robustness of GNN models. Moreover, TVGNN (Hansen & Bianchi, 2023) gradually distills the global label information from the node representations. Even though these models have achieved SOTA performance on the graph classification task, the imbalanced datasets bring a non-negligible problem for such models. Without specifically paying attention to the imbalanced nature of data, these SOTA graph classification models are not able to meet the requirements of the graph-level anomaly detection task, as we will show during the empirical evaluation.

## 3 Our Method: RQGNN

The observation in Section 1 highlights the differences between the Rayleigh Quotient distribution of anomalous and normal graphs. In Section 3.1, we further provide a theoretical analysis of the

Rayleigh Quotient. This motivates our design of the Rayleigh Quotient learning component (RQL) in our framework, to be elaborated in Section 3.2. Moreover, our theoretical analysis in Section 3.1 further shows that the accumulated energy of the graph can be represented by the Rayleigh Quotient, which motivates us to apply the spectral GNN to capture the spectral energy information, to be detailed in Section 3.3. We further present a powerful spectral-related pooling function called RQ-pooling in Section 3.3. Section 3.4 elaborates on the design of class-balanced focal loss.

## 3.1 RAYLEIGH QUOTIENT AND SPECTRAL ANALYSIS

As described in Section 1, Rayleigh Quotient has been studied in other domains. This inspires us to explore the property of the Rayleigh Quotient in the graph area. Specifically, we further provide analysis to show the connection between the Rayleigh Quotient and the spectrum of graphs. The following two theorems show that the change of the Rayleigh Quotient can be bounded given a small perturbation on graph signal $x$ and graph Laplacian $L$, and proofs can be found in Appendix A.1.

**Theorem 1.** *For any given graph $G$, if there exists a perturbation $\Delta$ on $L$, the change of Rayleigh Quotient can be bounded by $||\Delta||_2$.*

**Theorem 2.** *For any given graph $G$, if there exists a perturbation $\delta$ on $x$, the change of Rayleigh Quotient can be bounded by $2x^T L\delta + o(\delta)$. If $\delta$ is small enough, in which case $o(\delta)$ can be ignored, the change can be further bounded by $2x^T L\delta$.*

Theorems 1-2 provide valuable guidance in exploring the underlying spectral properties behind anomalous and normal graphs based on the Rayleigh Quotient. Recap from Section 1 that the normalized Rayleigh Quotient distribution of graphs with the same class label statistically exhibits a similar pattern on different sample sizes. If the graph Laplacian $L$ and graph signal $x$ of two graphs are close, then their Rayleigh Quotients will be close to each other and these two graphs will highly likely belong to the same class. This motivates us to design a component to learn the Rayleigh Quotient of each graph directly, as we will show in Section 3.2.

Besides, we further analyze the relationship between the Rayleigh Quotient and the spectral energy of graphs, which serves as the rationale for incorporating a spectral-related component into our framework. Let $\hat{x}_k^2 / \sum_{i=1}^{n} \hat{x}_i^2$ denote the spectral energy of $\lambda_k$. Although this distribution provides valuable guidance for measuring the graph spectrum in mathematics, it is not suitable for GNN training due to the time-consuming eigendecomposition computation. Therefore, in the following, we introduce the accumulated spectral energy and show that it can be transformed into the Rayleigh Quotient, thereby avoiding the expensive matrix decomposition process.

Let $\sum_{j=1}^{k} \hat{x}_j^2 / \sum_{i=1}^{n} \hat{x}_i^2$ denote the accumulated spectral energy from $\lambda_1$ to $\lambda_k$. According to previous work (Li et al., 2022; Luan et al., 2022), real-world graph data usually shows heterophily in connection and high-pass graph filters will capture more spectral information. Based on this observation, instead of exploring the original accumulated spectral energy that represents the low-frequency energy, we investigate the high-frequency energy that represents the accumulated spectral energy from $\lambda_k$ to $\lambda_n$. For any $t \in [\lambda_k, \lambda_{k+1})$, where $1 \leq k \leq n-1$, we denote $E(t) = 1 - \sum_{j=1}^{k} \hat{x}_j^2 / \sum_{i=1}^{n} \hat{x}_i^2$ as the high-frequency energy. Then we can derive:

$$\int_0^{\lambda_n} E(t)dt = \frac{\sum_{j=1}^{n} \lambda_j \hat{x}_j^2}{\sum_{i=1}^{n} \hat{x}_i^2} = \frac{x^T L x}{x^T x}. \tag{1}$$

This result demonstrates that the accumulated spectral energy can be exactly represented by the Rayleigh Quotient. We summarize this result in the following proposition.

**Proposition 1.** *Given graph $G$, the Rayleigh Quotient represents the accumulated spectral energy.*

The Proposition 1 indicates the Rayleigh Quotient represents the accumulated spectral energy of the graph, which motivates us to design a spectral GNN and spectral-related pooling function to capture the inherent properties behind anomalous graphs, as we will show in Section 3.3.

## 3.2 RAYLEIGH QUOTIENT LEARNING COMPONENT

Motivated by Theorems 1-2 and the observation in Section 1, a simple yet powerful component is introduced to capture different trends on the Rayleigh Quotient. Specifically, we first use a two-layer

MLP to obtain the latent representation of each node. Then, we calculate the Rayleigh Quotient for each graph as the explicit learning component in our RQGNN. Let $\tilde{\boldsymbol{X}}$ denote the node features $\boldsymbol{X}$ after the feature transformation, then the Rayleigh Quotient can be expressed as:

$$RQ(\boldsymbol{X}, \boldsymbol{L}) = diag\left(\frac{\tilde{\boldsymbol{X}}^T \boldsymbol{L} \tilde{\boldsymbol{X}}}{\tilde{\boldsymbol{X}}^T \tilde{\boldsymbol{X}}}\right), \tag{2}$$

where $diag(\cdot)$ denotes the diagonal entries of a square matrix. Finally, we employ another two-layer MLP to get the Rayleigh Quotient representation of the entire graph:

$$h_{RQ}^G = \text{MLP}\left(RQ(\boldsymbol{X}, \boldsymbol{L})\right). \tag{3}$$

Except for explicitly learning from the Rayleigh Quotient, following the common design of GNN, we need to implicitly learn from the topology and node features of graphs, so that we can collect comprehensive information for graph-level anomaly detection. The details are presented as follows.

### 3.3 Chebyshev wavelet GNN with RQ-pooling

As described in Section 3.1, the accumulated spectral energy can be represented by the Rayleigh Quotient, which reveals crucial spectral properties for graph-level anomaly detection. This motivates us to design a spectral GNN for learning graph representations. Even though existing spectral GNNs, e.g., ChebyNet (Defferrard et al., 2016) and BernNet (He et al., 2021), have achieved notable success in the node classification task, these models fall short in capturing the underlying properties of anomalous graphs, resulting in inferior performance on the graph-level anomaly detection task, as we will show in our experiments. This can be attributed to two main reasons. Firstly, simple spectral GNNs can be seen as single low-band or high-band graph filters (He et al., 2021), and each band filter only allows certain spectral energy to pass. However, as analyzed in Section 3.1, to capture the spectral properties of anomalous graphs, we should consider the spectral energy with respect to each eigenvalue. Therefore, it is necessary to employ multiple graph filters. The graph wavelet convolution model can be considered as a combination of different graph filters, enabling us to consider the spectral energy associated with each eigenvalue. Consequently, leveraging graph wavelet convolution provides advantages compared to using single graph filters. Secondly, even using more powerful spectral GNN models, generating improved graph representations remains challenging without a carefully designed pooling function. To address these issues, we present CWGNN-RQ, a novel component that effectively learns the inherent spectral representations of different graphs.

**CWGNN.** Following Hammond et al. (2011a), we define $\psi$ as the graph wavelet and a group of $q$ wavelets can be denoted as $\boldsymbol{W} = (\boldsymbol{W}_{\psi_1}, \boldsymbol{W}_{\psi_2}, \cdots, \boldsymbol{W}_{\psi_q})$. Each wavelet is denoted as $\boldsymbol{W}_{\psi_i} = \boldsymbol{U} g_i(\boldsymbol{\Lambda}) \boldsymbol{U}^T$, where $g_i(\cdot)$ is the kernel function defined on $[0, \lambda_n]$ in the spectral domain. Then, the general wavelet GNN of a graph signal $\boldsymbol{x}$ can be expressed as:

$$\boldsymbol{W}\boldsymbol{x} = \left[\boldsymbol{W}_{\psi_1}, \boldsymbol{W}_{\psi_2}, \cdots, \boldsymbol{W}_{\psi_q}\right] \boldsymbol{x} = \left[\boldsymbol{U} g_1(\boldsymbol{\Lambda}) \boldsymbol{U}^T \boldsymbol{x}, \boldsymbol{U} g_2(\boldsymbol{\Lambda}) \boldsymbol{U}^T \boldsymbol{x}, \cdots \boldsymbol{U} g_q(\boldsymbol{\Lambda}) \boldsymbol{U}^T \boldsymbol{x}\right].$$

However, calculating graph wavelets requires decomposing the graph Laplacian, resulting in expensive computational costs. To achieve better computational efficiency, we employ Chebyshev polynomials to calculate approximate wavelet operators. The following lemma shows that the Chebyshev series can be used to represent any function.

**Lemma 1** (Rivlin (1974)). *There always exists a convergent Chebyshev series for any function $f(t)$:*

$$f(t) = \frac{1}{2}c_0 + \sum_{k=1}^{\infty} c_k T_k(t),$$

*where $c_k = \frac{2}{\pi} \int_0^\pi cos(k\theta) f(cos(\theta)) d\theta$, and $k$ is the order of the Chebyshev polynomials.*

In addition, the Chebyshev polynomials on interval $[-1, 1]$ can be iteratively defined as $T_k(t) = 2t T_{k-1}(t) - T_{k-2}(t)$ with initial values of $T_0(t) = 1$ and $T_1(t) = t$. Given that the eigenvalues of the normalized Laplacian matrix fall in the range of $[0, 2]$, we utilize the shifted Laplacian matrix $\boldsymbol{L} - \boldsymbol{I}_n$ to compute the following shifted Chebyshev polynomials $\bar{T}$. Meanwhile, for each wavelet $i$, we also have a scale function $s_i(\cdot)$ to re-scale the eigenvalue so that it can fit into the domain of $f$ to calculate the following Chebyshev coefficient $\bar{c}_{i,k}$. By introducing the truncated Chebyshev

polynomials into graph wavelet operators, the $i$-th kernel function $f_i(\boldsymbol{L})$ is designed to capture the first $iK$-hop information, which can be expressed as follows (Hammond et al., 2011b):

$$f_i(\boldsymbol{L}) = \frac{1}{2}\bar{c}_{i,0}\boldsymbol{I}_n + \sum_{k=1}^{iK}\bar{c}_{i,k}\bar{T}_k(\boldsymbol{L}), \tag{4}$$

where $\bar{T}_k(\boldsymbol{L}) = \frac{4}{\lambda_n}(\boldsymbol{L}-\boldsymbol{I})\bar{T}_{k-1}(\boldsymbol{L}) - \bar{T}_{k-2}(\boldsymbol{L})$ with initial values of $\bar{T}_0(\boldsymbol{L}) = \boldsymbol{I}_n$ and $\bar{T}_1(\boldsymbol{L}) = t\boldsymbol{I}_n$ represents the shifted Chebyshev polynomials and $\bar{c}_{i,k} = \frac{2}{\pi}\int_0^\pi cos(k\theta)f(s_i(\frac{\lambda_n(cos(\theta)+1)}{2}))d\theta$ with $1 \leq i \leq q$. Then, the results of $q$ graph wavelets are concatenated together to generate the representation of node $j$:

$$\boldsymbol{h}_j = \text{CONCAT}\left(\left(f_1(\boldsymbol{L})\tilde{\boldsymbol{X}}\right)_j, \left(f_2(\boldsymbol{L})\tilde{\boldsymbol{X}}\right)_j, \cdots, \left(f_q(\boldsymbol{L})\tilde{\boldsymbol{X}}\right)_j\right). \tag{5}$$

After node representations $\boldsymbol{h}$ are generated by CWGNN, a pooling function is needed to obtain the representation of the entire graph. Commonly used pooling functions such as average pooling and sum pooling functions (Hamilton et al., 2017) have achieved satisfactory performance in various classification tasks. However, as demonstrated in our experiments, these techniques become ineffective in graph-level anomaly detection. Consequently, this challenge calls for a newly designed pooling function that can effectively guide CWGNN to learn better graph representation.

**RQ-pooling.** In order to incorporate spectral information into node weights, we adopt an attention mechanism to generate the graph representation:

$$\boldsymbol{h}_{Att}^G = \sigma\left(\sum_{j \in V} a_j\boldsymbol{h}_j\right), \tag{6}$$

where $\sigma$ is the non-linear activation function, and $a_j$ is the attention coefficient of node $j$. Specifically, since the spectral energy corresponds to each graph signal, we set the Rayleigh Quotient as the weight of these signals. Then, the attention coefficient for node $j$ can be expressed as $a_j = RQ(\boldsymbol{X}, \boldsymbol{L})\boldsymbol{h}_j$, which is used as the node importance score in RQ-pooling. Such a strategy allows CWGNN to capture more underlying spectral information of graphs. The final representation of the graph is the concatenation of both $\boldsymbol{h}_{RQ}^G$ and $\boldsymbol{h}_{Att}^G$:

$$\boldsymbol{h}^G = \text{MLP}\left(\text{CONCAT}\left(\boldsymbol{h}_{Att}^G, \boldsymbol{h}_{RQ}^G\right)\right). \tag{7}$$

### 3.4 CLASS-BALANCED FOCAL LOSS

As discussed in Section 2, the imbalanced nature of graph-level anomaly detection brings non-negligible challenges. To tackle this issue, we introduce a re-weighting technique called the class-balanced focal loss, which enhances the anomalous detection capability of our RQGNN.

**Expected number** (Cui et al., 2019). As the number of training samples increases, there will be more potential information overlap among different samples. Consequently, the marginal benefit that a model can extract from the data diminishes. To address this issue, the class-balanced focal loss is designed to capture the diminishing marginal benefits by using more data points of a class. This approach ensures that the model effectively utilizes the available data while avoiding redundancy and maximizing its learning potential. Specifically, we define the expected number $\eta(n_t)$ as the total number of samples that can be covered by $n_t$ training data and utilize the inverse of this number as the balance factor in our loss function.

**Proposition 2.** *The expected number* $\eta(n_t) = \frac{1-\beta^{n_t}}{1-\beta}$, *where* $\beta = \frac{N-1}{N}$ *with* $N$ *equaling to the total number of data points in class* $t$.

In practice, without further information of data for each class, it is difficult to empirically find a set of good $N$ for all classes. Therefore, we assume $N$ is dataset-dependent and set the same $\beta$ for all classes in a dataset. In addition, we also employ focal loss (Lin et al., 2017), an adjusted version of cross-entropy loss. By combining the expected number and focal loss together, we can achieve the goal of reweighting the loss for each class. The class-balanced focal loss is defined as follows:

$$\mathcal{L}_{CB_{focal}} = \frac{\mathcal{L}_{focal}}{\eta(n_y)} = \frac{1-\beta}{1-\beta^{n_y}}\sum_{i=1}^C (1-p_i)^\gamma log(p_i), \tag{8}$$

Table 1: AUC and Macro-F1 scores (%) on ten datasets with random split.

| Datasets | Metrics | Spectral GNN | | Graph Classification | | | Graph-level Anomaly Detection | | | | | | | |
|---|---|---|---|---|---|---|---|---|---|---|---|---|---|---|
| | | ChebyNet | BernNet | GMT | Gmixup | TVGNN | OCGIN | OCGTL | GLocalKD | HimNet | iGAD | RQGNN-1 | RQGNN-2 | RQGNN |
| MCF-7 | AUC | 0.6612 | 0.6172 | 0.7706 | 0.6954 | 0.7180 | 0.5348 | 0.5866 | 0.6363 | 0.6369 | 0.8146 | 0.8094 | 0.8346 | **0.8354** |
| | F1 | 0.4780 | 0.4784 | 0.4784 | 0.4779 | 0.5594 | - | - | - | - | 0.6468 | 0.6626 | 0.7205 | **0.7394** |
| MOLT-4 | AUC | 0.6647 | 0.6144 | 0.7606 | 0.6232 | 0.7159 | 0.5299 | 0.6191 | 0.6631 | 0.6633 | 0.8086 | 0.8246 | 0.8196 | **0.8316** |
| | F1 | 0.4854 | 0.4794 | 0.4814 | 0.4789 | 0.4916 | - | - | - | - | 0.6617 | 0.7119 | 0.7113 | **0.7240** |
| PC-3 | AUC | 0.6051 | 0.6094 | 0.7896 | 0.6908 | 0.7974 | 0.5810 | 0.6349 | 0.6727 | 0.6703 | 0.8723 | 0.8553 | 0.8671 | **0.8782** |
| | F1 | 0.4853 | 0.4853 | 0.4853 | 0.4852 | 0.6206 | - | - | - | - | 0.6697 | 0.7003 | **0.7241** | 0.7184 |
| SW-620 | AUC | 0.6759 | 0.6072 | 0.7467 | 0.6479 | 0.7326 | 0.4955 | 0.6398 | 0.6542 | 0.6544 | 0.8512 | 0.8401 | 0.8427 | **0.8550** |
| | F1 | 0.4898 | 0.4847 | 0.4874 | 0.4844 | 0.5365 | - | - | - | - | 0.6627 | 0.6941 | 0.7209 | **0.7335** |
| NCI-H23 | AUC | 0.6728 | 0.6114 | 0.8030 | 0.7324 | 0.7782 | 0.4948 | 0.6122 | 0.6837 | 0.6814 | 0.8297 | 0.8413 | 0.8554 | **0.8680** |
| | F1 | 0.4930 | 0.4869 | 0.4869 | 0.4869 | 0.5520 | - | - | - | - | 0.6646 | 0.6735 | **0.7349** | 0.7214 |
| OVCAR-8 | AUC | 0.6303 | 0.5850 | 0.7692 | 0.5663 | 0.7653 | 0.5298 | 0.6007 | 0.6750 | 0.6757 | 0.8691 | 0.8549 | 0.8650 | **0.8799** |
| | F1 | 0.4900 | 0.4868 | 0.4868 | 0.4869 | 0.5406 | - | - | - | - | 0.6638 | 0.6876 | 0.7077 | **0.7215** |
| P388 | AUC | 0.7266 | 0.6707 | 0.8495 | 0.6516 | 0.7957 | 0.5252 | 0.6501 | 0.6445 | 0.6667 | 0.8995 | 0.8911 | 0.8904 | **0.9023** |
| | F1 | 0.5635 | 0.5001 | 0.6583 | 0.4856 | 0.5557 | - | - | - | - | 0.7437 | 0.7552 | 0.7738 | **0.7963** |
| SF-295 | AUC | 0.6650 | 0.6353 | 0.7992 | 0.6471 | 0.7346 | 0.4774 | 0.6440 | 0.7069 | 0.7073 | 0.8770 | 0.8691 | 0.8781 | **0.8825** |
| | F1 | 0.4871 | 0.4871 | 0.4871 | 0.4866 | 0.4935 | - | - | - | - | 0.6919 | 0.7120 | 0.7335 | **0.7416** |
| SN12C | AUC | 0.6598 | 0.6014 | 0.7919 | 0.7211 | 0.7441 | 0.5004 | 0.5617 | 0.6880 | 0.6916 | 0.8747 | 0.8851 | **0.8904** | 0.8861 |
| | F1 | 0.4972 | 0.4874 | 0.4874 | 0.4871 | 0.5437 | - | - | - | - | 0.6714 | 0.7204 | 0.7549 | **0.7660** |
| UACC257 | AUC | 0.6584 | 0.6115 | 0.7735 | 0.6564 | 0.7410 | 0.5411 | 0.6148 | 0.6647 | 0.6659 | 0.8512 | 0.8447 | 0.8596 | **0.8724** |
| | F1 | 0.4894 | 0.4895 | 0.4894 | 0.4904 | 0.5373 | - | - | - | - | 0.6483 | 0.6889 | 0.7087 | **0.7362** |

where $\beta$ and $\gamma$ are hyperparameters, $n_y$ is the number of samples in class $y$ that the current sample belongs to, $C$ denotes the number of classes, and $p_i = \mathrm{softmax}(\boldsymbol{h}^G)_i$ is the predicted probability for the current sample belonging to class $i$.

# 4 EXPERIMENTS

## 4.1 EXPERIMENTAL SETUP

**Datasets.** We use 10 real-world datasets to investigate the performance of RQGNN, including MCF-7, MOLT-4, PC-3, SW-620, NCI-H23, OVCAR-8, P388, SF-295, SN12C, and UACC257. These datasets are obtained from the TUDataset (Morris et al., 2020), consisting of various chemical compounds and their reactions to different cancer cells. We treat inactive chemical compounds as normal graphs and active ones as anomalous graphs.

**Baselines.** We compare RQGNN against 10 SOTA GNN competitors, including spectral GNNs, graph classification models and graph-level anomaly detection models.

- Spectral GNNs with average pooling function: ChebyNet (Defferrard et al., 2016) and BernNet (He et al., 2021).
- Graph classification models: GMT (Baek et al., 2021), Gmixup (Han et al., 2022), and TVGNN (Hansen & Bianchi, 2023).
- Graph-level anomaly detection models: OCGIN (Zhao & Akoglu, 2021), OCGTL (Qiu et al., 2022), GLocalKD (Ma et al., 2022), HimNet (Niu et al., 2023), and iGAD (Zhang et al., 2022).

Also, we investigate two variants of RQGNN. We use RQGNN-1 to indicate the model that replaces the RQ-pooling with the average pooling, and use RQGNN-2 to indicate the model without the RQL component. More details of the datasets and baselines can be found in Appendix A.3.

**Experimental Settings.** We randomly divide each dataset into training/validation/test sets with 70%/15%/15%, respectively. During the sampling process, we ensure that each set maintains a consistent ratio between normal and anomalous graphs. We select the epoch where the models achieve the best Macro-F1 score on the validation set as the best epoch and use the corresponding model for performance evaluation. We set the learning rate as 0.005, the batch size as 512, the hidden dimension $d = 64$, the width of CWGNN-RQ $q = 4$, the depth of CWGNN-RQ $K = 6$, the dropout rate as 0.4, the hyperparameters of the loss function $\beta = 0.999$, $\gamma = 1.5$, and we use batch normalization for the final graph embeddings. We obtain the source code of all competitors from GitHub and perform these GNN models with default parameter settings suggested by their authors.

## 4.2 EXPERIMENTAL RESULTS

We first evaluate the performance of RQGNN against different SOTA GNN models. Table 1 reports AUC and Macro-F1 scores of each GNN model on 10 datasets. The best result on each dataset is

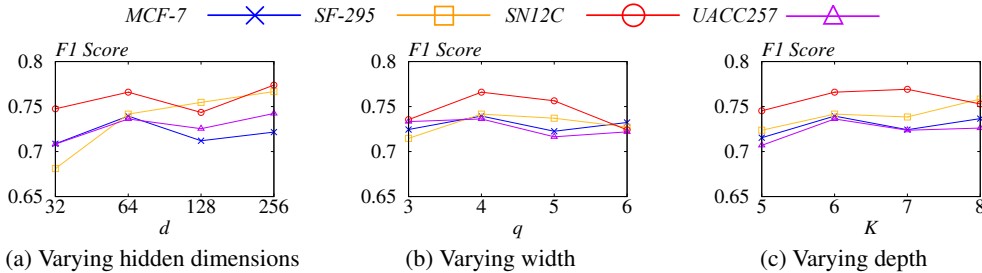

(a) Varying hidden dimensions      (b) Varying width      (c) Varying depth

Figure 2: Varying the hidden dimension, width, and depth.

highlighted in boldface. Since OCGIN, OCGTL, GLocalKD, and HimNet adopt one-class classification to identify anomalous graphs, we only report their AUC scores. As we can see, RQGNN outperforms all baselines on all datasets. Next, we provide our detailed observations.

Firstly, for two SOTA spectral GNNs, ChebyNet and BernNet, they fail to learn the underlying anomalous properties from the spectral perspective. In particular, compared with ChebyNet and BernNet, RQGNN takes the lead by 20.72% and 25.28% on these 10 datasets in terms of average AUC score, and takes the lead by 24.40% and 25.33% on these 10 datasets in terms of average Macro-F1 score, respectively. These empirical evidences demonstrate that even though graph Laplacian matrix is related to graph spectral energy, we still need to carefully design graph filters and pooling functions to capture the underlying anomalous properties in the graph.

Secondly, we carefully check the results of GNNs for graph classification to verify whether graph-level anomaly detection can be easily tackled by graph classification models. From Table 1, we can observe that compared to three GNN models, GMT, Gmixup, and TVGNN, RQGNN takes a lead by 8.38%, 20.59%, and 11.69% in terms of average AUC score and 23.70%, 25.48%, and 19.67% in terms of average Macro-F1 score, respectively. These results demonstrate that graph classification models cannot be directly adopted to handle the graph-level anomaly detection task.

Thirdly, we compare RQGNN with SOTA GNN models designed for graph-level anomaly detection. Despite being specialized for graph-level anomaly detection, OCGIN, OCGTL, GLocalKD, and HimNet fail to outperform other GNN baselines in terms of AUC scores. This can be attributed to their inability to effectively capture the important graph anomalous information. In contrast, RQGNN guided by the Rayleigh Quotient successfully captures the spectral differences between anomalous and normal graphs, resulting in significantly superior performance compared to OCGIN, OCGTL, GLocalKD, and HimNet. In particular, RQGNN takes the lead by an average margin of 34.82%, 25.28%, 20.02%, and 19.78% in terms of AUC score, respectively. Among all the baselines, iGAD stands out as the most competitive model, which incorporates anomalous-aware sub-structural information into node representations. However, it lacks the incorporation of important properties of anomalous graphs, such as the Rayleigh Quotient and spectral energy of graphs, which leads to relatively unsatisfying Macro-F1 scores on all datasets. With the guidance of Rayleigh Quotient, RQGNN outperforms iGAD by 1.44% in terms of AUC score and 6.74% in terms of Macro-F1 score on average across 10 datasets.

## 4.3 ABLATION STUDY

In this set of experiments, we investigate the effectiveness of each component in RQGNN. Ablation study for the class-balanced focal loss can be found in Appendix A.2. The experimental results of RQGNN variants are shown in Table 1.

Firstly, we use RQGNN-1 to indicate the model that replaces the RQ-pooling with average pooling. Recap from Section 4.1 that, it combines the representation of Rayleigh Quotient and CWGNN with average pooling function. As we can observe, RQGNN-1 outperforms all other baselines on all datasets in terms of the Macro-F1 scores. In particular, compared with RQGNN-1, RQGNN further boosts the performance and takes the lead by 1.76% in terms of the AUC score and 3.92% in terms of the Macro-F1 score on average. This result demonstrates that the RQ-pooling that introduces Rayleigh Quotient as the node weight captures more crucial information from the spectral domain.

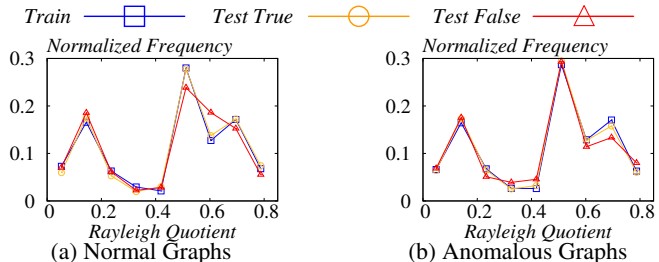

Figure 3: Normalized Rayleigh Quotient distribution on SN12C.

Then, we use RQGNN-2 to indicate the model with only CWGNN-RQ. Specifically, we remove the RQL component that explicitly calculates the Rayleigh Quotient of each graph. Instead, we only compute the Rayleigh Quotient as the node weights for CWGNN. As we can observe, RQGNN-2 outperforms all the other baselines in terms of the Macro-F1 scores, which again shows the effectiveness of the RQ-pooling. Besides, according to Table 1, we can see that RQGNN is 0.09% higher than RQGNN-2 in terms of AUC score and 1.08% higher than RQGNN-2 in terms of Macro-F1 score on average, which further shows the effectiveness of the RQL component. In summary, these results demonstrate the effectiveness of each component in RQGNN.

### 4.4 PARAMETER ANALYSIS

Next, we conduct experiments to analyze the effect of representative parameters: the hidden dimension $d$ of RQGNN, the width $q$ and depth $K$ of CWGNN-RQ on MCF-7, SF-295, SN12C, and UACC257 datasets. Figure 2 reports the Macro-F1 score of RQGNN as we vary the hidden dimension $d$ from $32$ to $256$, the width $q$ from $3$ to $6$, and the depth $K$ from $5$ to $8$. As we can observe, when we set the hidden dimension to $64$, RQGNN achieves relatively satisfactory performances on these four datasets. When the width of CWGNN-RQ is set to $4$, RQGNN achieves the best results on all four datasets. Hence, we set the width to $4$ in our experiments. Meanwhile, as we can observe, RQGNN shows a relatively stable and high performance in terms of all four presented datasets when we set the depth to $6$. As a result, the depth is set to $6$ in RQGNN.

### 4.5 CASE STUDY

In this set of experiments, we conduct experiments to investigate whether RQGNN learns the trends of Rayleigh Quotient on anomalous and normal graphs. If RQGNN successfully detects anomalous graphs, the samples in the test set that can be classified correctly by a converged model should have a similar Rayleigh Quotient distribution to that in the training set. Figure 3 illustrates the Rayleigh Quotient distribution of the normal and anomalous graphs in the training and test set on the SN12C dataset. As we can see, graphs that can be classified correctly in the test set exhibit a similar Rayleigh Quotient distribution to that in the training set. Meanwhile, those graphs that our RQGNN can not classify correctly, display different distributions from the train graphs. These results demonstrate that Rayleigh Quotient is an intrinsic characteristic of the graph-level anomaly detection task for the application studied. Our RQGNN can effectively learn the Rayleigh Quotient as a discriminative feature and thus outperforms SOTA competitors.

## 5 CONCLUSION

In this paper, we introduce spectral analysis into the graph-level anomaly detection task. We discover differences in the spectral energy distributions between anomalous and normal graphs and further demonstrate the observation through comprehensive experiments and theoretical analysis. The combination of the RQL component that explicitly captures the Rayleigh Quotient of the graph and CWGNN-RQ that implicitly explores graph anomalous information provides different spectral perspectives for this task. Extensive experiments demonstrate that RQGNN consistently outperforms other SOTA competitors by a significant margin.

ACKNOWLEDGMENTS

This work was supported by Hong Kong RGC GRF (Grant No. 14217322) and Hong Kong ITC ITF (Grant No.MRP/071/20X). Sibo Wang is the corresponding author.

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

# A APPENDIX

## A.1 PROOFS

**Proof of Theorem 1.** The following lemma is used for the proof.

**Lemma 2** (Courant-Fischer Theorem). *Let $M \in \mathbb{R}^{n \times n}$ be a symmetric matrix with eigenvalues $\mu_1 \leq \mu_2 \leq \cdots \leq \mu_n$. Then,*

$$\mu_k = \max_{S \subseteq \mathbb{R}^n, dim(s)=k} \min_{\boldsymbol{x} \in S, \boldsymbol{x} \neq \boldsymbol{0}} \frac{\boldsymbol{x}^T \boldsymbol{M} \boldsymbol{x}}{\boldsymbol{x}^T \boldsymbol{x}} = \min_{T \subseteq \mathbb{R}^n, dim(T)=n-k+1} \max_{\boldsymbol{x} \in T, \boldsymbol{x} \neq \boldsymbol{0}} \frac{\boldsymbol{x}^T \boldsymbol{M} \boldsymbol{x}}{\boldsymbol{x}^T \boldsymbol{x}},$$

*where the maximization and minimization are over subspaces $S$ and $T$ of $\mathbb{R}^n$.*

Let $\lambda_i(\cdot)$ denote the $i$-th eigenvalue of the input matrix, where the eigenvalue is in ascending order according to the index. For symmetric matrix $\boldsymbol{L}$ and $\boldsymbol{\Delta}$, according to Lemma 2, we have:

$$\lambda_i(\boldsymbol{L} + \boldsymbol{\Delta}) = \min_{T \subseteq \mathbb{R}^n, dim(T)=n-i+1} \max_{\boldsymbol{x} \in T, \boldsymbol{x} \neq \boldsymbol{0}} \frac{\boldsymbol{x}^T(\boldsymbol{L} + \boldsymbol{\Delta})\boldsymbol{x}}{\boldsymbol{x}^T \boldsymbol{x}}$$

$$= \min_{T \subseteq \mathbb{R}^n, dim(T)=n-i+1} (\max_{\boldsymbol{x} \in T, \boldsymbol{x} \neq \boldsymbol{0}} \frac{\boldsymbol{x}^T \boldsymbol{L} \boldsymbol{x}}{\boldsymbol{x}^T \boldsymbol{x}} + \max_{\boldsymbol{x} \in T, \boldsymbol{x} \neq \boldsymbol{0}} \frac{\boldsymbol{x}^T \boldsymbol{\Delta} \boldsymbol{x}}{\boldsymbol{x}^T \boldsymbol{x}})$$

$$\leq \min_{T \subseteq \mathbb{R}^n, dim(T)=n-i+1} (\max_{\boldsymbol{x} \in T, \boldsymbol{x} \neq \boldsymbol{0}} \frac{\boldsymbol{x}^T \boldsymbol{L} \boldsymbol{x}}{\boldsymbol{x}^T \boldsymbol{x}} + \lambda_n(\boldsymbol{\Delta}))$$

$$= \lambda_i(\boldsymbol{L}) + \lambda_n(\boldsymbol{\Delta}).$$

Similarly, by employing the other format of Lemma 2, we can derive $\lambda_i(\boldsymbol{L}) + \lambda_1(\boldsymbol{\Delta}) \leq \lambda_i(\boldsymbol{L}+\boldsymbol{\Delta})$, from which it follows that

$$\max_i \{|\lambda_i(\boldsymbol{L} + \boldsymbol{\Delta}) - \lambda_i(\boldsymbol{L})|\} \leq \max\{|\lambda_n(\boldsymbol{\Delta})|, |\lambda_1(\boldsymbol{\Delta})|\} = ||\boldsymbol{\Delta}||_2.$$

Hence, we can rewrite the inequality as:

$$\max_i \{|\lambda_i(\boldsymbol{L}) - \lambda_i(\boldsymbol{L} + \boldsymbol{\Delta})|\} \leq ||\boldsymbol{\Delta}||_2.$$

Considering the change of Rayleigh Quotient, we have:

$$\frac{\boldsymbol{x}^T(\boldsymbol{L} + \boldsymbol{\Delta})\boldsymbol{x}}{\boldsymbol{x}^T \boldsymbol{x}} - \frac{\boldsymbol{x}^T \boldsymbol{L} \boldsymbol{x}}{\boldsymbol{x}^T \boldsymbol{x}} = \frac{\sum_{j=1}^n \lambda_j(\boldsymbol{L} + \boldsymbol{\Delta})\hat{x}_j^2}{\sum_{i=1}^n \hat{x}_i^2} - \frac{\sum_{j=1}^n \lambda_j(\boldsymbol{L})\hat{x}_j^2}{\sum_{i=1}^n \hat{x}_i^2}$$

$$= \frac{\sum_{j=1}^n (\lambda_j(\boldsymbol{L} + \boldsymbol{\Delta}) - \lambda_j(\boldsymbol{L}))\hat{x}_j^2}{\sum_{i=1}^n \hat{x}_i^2}$$

$$\leq \frac{\max_t \left\{ (\lambda_t(\boldsymbol{L} + \boldsymbol{\Delta}) - \lambda_t(\boldsymbol{L})) \sum_{j=1}^n \hat{x}_j^2 \right\}}{\sum_{i=1}^n \hat{x}_i^2}$$

$$\leq \max_t (\lambda_t(\boldsymbol{L} + \boldsymbol{\Delta}) - \lambda_t(\boldsymbol{L})) \leq ||\boldsymbol{\Delta}||_2. \square$$

**Proof of Theorem 2.** For simplicity, let $\boldsymbol{x}$ be a normalized vector so that the Rayleigh Quotient has the form as $\boldsymbol{x}^T \boldsymbol{L} \boldsymbol{x}$. Then, we have:

$$(\boldsymbol{x} + \boldsymbol{\delta})^T \boldsymbol{L}(\boldsymbol{x} + \boldsymbol{\delta}) - \boldsymbol{x}^T \boldsymbol{L} \boldsymbol{x}$$

$$= \boldsymbol{x}^T \boldsymbol{L} \boldsymbol{x} + \boldsymbol{\delta}^T \boldsymbol{L} \boldsymbol{x} + \boldsymbol{x}^T \boldsymbol{A} \boldsymbol{\delta} + \boldsymbol{\delta}^T \boldsymbol{L} \boldsymbol{\delta} - \boldsymbol{x}^T \boldsymbol{L} \boldsymbol{x}$$

$$= 2 \boldsymbol{x}^T \boldsymbol{L} \boldsymbol{\delta} + o(\boldsymbol{\delta})$$

If $o(\boldsymbol{\delta})$ is small enough, in which case $o(\boldsymbol{\delta})$ can be ignored, the change is bounded by $2\boldsymbol{x}^T \boldsymbol{L} \boldsymbol{\delta}$. $\square$

**Proof of Proposition 2.** The result can be derived by mathematical induction. Let $n = 1$ be the initial case. It is obvious that $\eta(1) = \frac{1-\beta^1}{1-\beta} = 1$, so $n = 1$ holds. Assume that this proposition holds for $n = k - 1$, and the probability of sampling $k$-th data is $p = \eta(k-1)/N$. Therefore, the expected number after sampling $k$-th data is:

$$\eta(k) = p\eta(k-1) + (1-p)(\eta(k-1) + 1) = 1 + \frac{N-1}{N}\eta(k-1)$$

Table 2: Ablation study for class-balanced focal loss.

| Datasets | Metrics | ChebyNet | ChebyNet-CB | BernNet | BernNet-CB | TVGNN | TVGNN-CB | Gmixup | Gmixup-CB | GMT | GMT-CB | RQGNN-3 | RQGNN |
|---|---|---|---|---|---|---|---|---|---|---|---|---|---|
| MCF-7 | AUC | 0.6612 | 0.6617 | 0.6172 | 0.6215 | 0.7180 | 0.7309 | 0.6954 | 0.6984 | 0.7706 | 0.7479 | 0.8239 | 0.8354 |
| | F1 | 0.4780 | 0.4804 | 0.4784 | 0.4784 | 0.5594 | 0.5608 | 0.4779 | 0.4779 | 0.4784 | 0.4784 | 0.7293 | 0.7394 |
| MOLT-4 | AUC | 0.6647 | 0.6650 | 0.6144 | 0.6068 | 0.7159 | 0.7574 | 0.6232 | 0.6644 | 0.7606 | 0.6226 | 0.8341 | 0.8316 |
| | F1 | 0.4854 | 0.4875 | 0.4794 | 0.4794 | 0.4916 | 0.5793 | 0.4789 | 0.4789 | 0.4814 | 0.4794 | 0.7213 | 0.7240 |
| PC-3 | AUC | 0.6051 | 0.6065 | 0.6094 | 0.6040 | 0.7974 | 0.7782 | 0.6908 | 0.6592 | 0.7896 | 0.7877 | 0.8520 | 0.8782 |
| | F1 | 0.4853 | 0.4853 | 0.4853 | 0.4853 | 0.6206 | 0.6136 | 0.4852 | 0.4852 | 0.4853 | 0.4853 | 0.7287 | 0.7184 |
| SW-620 | AUC | 0.6759 | 0.6823 | 0.6072 | 0.6081 | 0.7326 | 0.7332 | 0.6479 | 0.6202 | 0.7467 | 0.7620 | 0.8504 | 0.8550 |
| | F1 | 0.4898 | 0.4947 | 0.4847 | 0.4847 | 0.5365 | 0.5547 | 0.4844 | 0.4844 | 0.4874 | 0.4847 | 0.7081 | 0.7335 |
| NCI-H23 | AUC | 0.6728 | 0.6734 | 0.6114 | 0.6289 | 0.7782 | 0.7810 | 0.7324 | 0.6793 | 0.8030 | 0.7808 | 0.8539 | 0.8680 |
| | F1 | 0.4930 | 0.5203 | 0.4869 | 0.4869 | 0.5520 | 0.5484 | 0.4869 | 0.4869 | 0.4869 | 0.4869 | 0.7340 | 0.7214 |
| OVCAR-8 | AUC | 0.6303 | 0.6294 | 0.5850 | 0.5803 | 0.7653 | 0.7730 | 0.5663 | 0.6591 | 0.7692 | 0.7713 | 0.8563 | 0.8799 |
| | F1 | 0.4900 | 0.4992 | 0.4868 | 0.4868 | 0.5406 | 0.5938 | 0.4869 | 0.4869 | 0.4868 | 0.4868 | 0.7126 | 0.7215 |
| P388 | AUC | 0.7266 | 0.7324 | 0.6707 | 0.6694 | 0.7957 | 0.7757 | 0.6516 | 0.6562 | 0.8495 | 0.8638 | 0.8883 | 0.9023 |
| | F1 | 0.5635 | 0.5656 | 0.5001 | 0.5002 | 0.5557 | 0.5549 | 0.4856 | 0.4856 | 0.6583 | 0.6268 | 0.7897 | 0.7963 |
| SF-295 | AUC | 0.6650 | 0.6670 | 0.6353 | 0.6302 | 0.7346 | 0.7722 | 0.6471 | 0.7070 | 0.7992 | 0.8004 | 0.8792 | 0.8825 |
| | F1 | 0.4871 | 0.4871 | 0.4871 | 0.4871 | 0.4935 | 0.5779 | 0.4866 | 0.4866 | 0.4871 | 0.4871 | 0.7074 | 0.7416 |
| SN12C | AUC | 0.6598 | 0.6634 | 0.6014 | 0.5992 | 0.7441 | 0.7710 | 0.7211 | 0.7017 | 0.7919 | 0.7985 | 0.8803 | 0.8861 |
| | F1 | 0.4972 | 0.4970 | 0.4874 | 0.4874 | 0.5437 | 0.5736 | 0.4871 | 0.4871 | 0.4874 | 0.4875 | 0.7554 | 0.7660 |
| UACC257 | AUC | 0.6584 | 0.6645 | 0.6115 | 0.6130 | 0.7410 | 0.7450 | 0.6564 | 0.6097 | 0.7735 | 0.7845 | 0.8537 | 0.8724 |
| | F1 | 0.4894 | 0.4933 | 0.4895 | 0.4895 | 0.5373 | 0.5330 | 0.4904 | 0.4904 | 0.4894 | 0.4895 | 0.7095 | 0.7362 |

Table 3: Tumor description of the 10 datasets

| Dataset | MCF-7 | MOLT-4 | PC-3 | SW-620 | NCI-H23 | OVCAR-8 | P388 | SF-295 | SN12C | UACC-257 |
|---|---|---|---|---|---|---|---|---|---|---|
| Tumor | Breast | Leukemia | Prostate | Colon | Non-Small Cell Lung | Ovarian | Leukemia | Central Nervous System | Rental | Melanoma |

Since we assume that $\eta(k-1) = \frac{1-\beta^{k-1}}{1-\beta}$, so we can derive:

$$\eta(k) = 1 + \beta\frac{1-\beta^{k-1}}{1-\beta} = \frac{1-\beta^k}{1-\beta}$$

which proves the expected number of samples is $\eta(k) = \frac{1-\beta^k}{1-\beta}$. □

## A.2 ABLATION STUDY FOR CLASS-BALANCED FOCAL LOSS

In this set of experiments, we explore the impact of replacing the cross-entropy loss function with the class-balanced focal loss in spectral GNN, i.e.,ChebyNet (Defferrard et al., 2016) and BernNet (He et al., 2021) and GNN models designed for graph classification tasks, i.e., TVGNN-CB (Hansen & Bianchi, 2023), Gmixup-CB (Han et al., 2022), and GMT-CB (Baek et al., 2021). In addition, we introduce RQGNN-3 as a variant of RQGNN that uses the cross-entropy loss instead of the class-balanced focal loss during training. The results of these models are summarized in Table 2. As we can see, the class-balanced focal loss does not yield benefits for ChebyNet, BernNet, BG-mixup, and GMT. This further demonstrates that even though tackling the imbalanced problem of datasets, these spectral GNNs and graph classification models fail to capture the underlying properties of anomalous graphs. For TVGNN, class-balanced focal loss improves the performance to some extent. However, it still falls way behind models specifically designed for graph-level anomaly detection. This observation highlights the differences between graph classification and graph-level anomaly detection tasks. When comparing RQGNN-3 with RQGNN, we notice the performance drops on almost all datasets. This indicates that the class-balanced focal loss can indeed benefit graph anomaly detection models.

## A.3 BASELINES AND DATASETS

**Baselines.** The first group is Spectral GNNs:

- ChebyNet (Defferrard et al., 2016): a GNN with Chebyshev polynomials as convolution kernels;
- BernNet (He et al., 2021): a GNN which utilizes Bernstein approximation of spectral filters.

The second group is GNN models designed for graph classification:

- GMT (Baek et al., 2021): a GNN with multi-head attention-based global pooling layer to capture the interactions between nodes and topology of graphs;
- Gmixup (Han et al., 2022): a GNN with data augmentation to improve the robustness;
- TVGNN (Hansen & Bianchi, 2023): a GNN with knowledge distillation of node representations.

The third group is SOTA GNN models for graph-level anomaly detection:

Table 4: Statistics of 10 real-world datasets, where $n_n$ is the number of normal graphs, $n_a$ is the number of anomalous graphs, $h = \frac{n_a}{n_n+n_a}$ is the anomalous ratio, $\bar{n}$ is the average number of nodes, $\bar{m}$ is the average number of edges, and $F$ is the number of attributes.

| Dataset | MCF-7 | MOLT-4 | PC-3 | SW-620 | NCI-H23 | OVCAR-8 | P388 | SF-295 | SN12C | UACC257 |
|---|---|---|---|---|---|---|---|---|---|---|
| $n_n$ | 25476 | 36625 | 25941 | 38122 | 38296 | 38437 | 39174 | 38246 | 38049 | 38345 |
| $n_a$ | 2294 | 3140 | 1568 | 2410 | 2057 | 2079 | 2298 | 2025 | 1955 | 1643 |
| $h$ | 0.0826 | 0.079 | 0.057 | 0.0595 | 0.051 | 0.0513 | 0.0554 | 0.0503 | 0.0489 | 0.0411 |
| $\bar{n}$ | 26.4 | 26.1 | 26.36 | 26.06 | 26.07 | 26.08 | 22.11 | 26.06 | 26.08 | 262.09 |
| $\bar{m}$ | 28.53 | 28.14 | 28.49 | 28.09 | 28.1 | 28.11 | 23.56 | 28.09 | 28.11 | 28.13 |
| $F$ | 46 | 64 | 45 | 65 | 65 | 65 | 72 | 65 | 65 | 64 |

- OCGIN (Zhao & Akoglu, 2021): the first GNN to explore graph-level anomaly detection;
- OCGTL (Qiu et al., 2022): a GNN with graph transformation learning;
- GLocalKD (Ma et al., 2022): a GNN with the random distillation technique;
- HimNet (Niu et al., 2023): a GNN with a hierarchical memory framework to balance the anomaly-related local and global information;
- iGAD (Zhang et al., 2022): a GNN with a substructure-aware component to capture topological features and a node-aware component to capture node features.

**Datasets.** The datasets used in our experiments are collected from PubChem by TUDataset (Morris et al., 2020). According to PubChem, these 10 datasets provide information on the biological activities of small molecules, where nodes denote atoms in the chemical compounds, and edges represent the chemical bonds between two atoms. These datasets contain the bioassay records for anticancer screen tests with different cancer cell lines. Each dataset belongs to a certain type of cancer screen with the outcome active or inactive. We treat inactive chemical compounds as normal graphs and active ones as anomalous graphs. In addition, the attributes are generated from node labels using one-hot encoding.

The statistics of these 10 real-world datasets are shown in Table4 and Table 3 provides a brief tumor description of these datasets.

## A.4 DATASETS BEYOND BIOGRAPHY DOMAIN

Table 5: Datasets beyond the biography domain

| Datasets | Metrics | ChebyNet | BernNet | TVGNN | GMT | HimNet | iGAD | RQGNN |
|---|---|---|---|---|---|---|---|---|
| COLORS-3 | AUC | 0.6192 | 0.6136 | 0.5000 | 0.5063 | 0.5129 | 0.7385 | 0.9378 |
| | F1 | 0.4764 | 0.4764 | 0.4764 | 0.4764 | 0.5934 | 0.5301 | 0.7640 |
| DBLP_v1 | AUC | 0.8369 | 0.8549 | 0.8455 | 0.8234 | 0.6656 | 0.7346 | 0.8808 |
| | F1 | 0.5472 | 0.4877 | 0.6449 | 0.4877 | 0.6133 | 0.6648 | 0.7709 |
| COLORS-3-ind | AUC | 0.6500 | 0.6486 | 0.5000 | 0.5000 | 0.4522 | 0.7630 | 0.9421 |
| | F1 | 0.3710 | 0.3710 | 0.3710 | 0.3710 | 0.4707 | 0.6674 | 0.8423 |

Although the main application of graph-level anomaly detection is typically focused on detecting chemical compounds, there are still applications in other domains such as social networks. However, due to the lack of datasets specifically collected for graph-level anomaly detection in other domains, we construct new datasets and conduct experiments on them. To be comprehensive, we use two graph classification datasets to evaluate our RQGNN, one is COLORS-3 (Knyazev et al., 2019), which is a synthetic dataset in TUDataset repository (Morris et al., 2020), and the other is DBLP_v1 (Pan et al., 2013) which is a social network dataset as classified in TUDataset.

Specifically, we first conduct experiments on the COLORS-3 dataset, which contains 11 classes. We use 10 classes to form normal graphs and use the remaining class as anomalous graphs.

Then, we conduct additional experiments in the social network domain using the DBLP_v1 dataset, which is a well-balanced social network classification dataset in the field of computer science. In this dataset, each graph denotes a research paper belonging to either DBDM (database and data mining) or CVPR (computer vision and pattern recognition) field, where each node denotes a paper ID or a keyword and each edge denotes the citation relationship between papers or keyword relations in the

title. To create a new graph-level anomaly detection dataset, we randomly sample 5% of one class as the anomaly class and use the other class as the normal class.

In addition, despite our primary focus not being on the out-of-distribution task, we still conduct additional experiments to evaluate the performance of our proposed model. We use a modified COLORS-3 dataset to construct COLORS-3-ind dataset. We randomly select two classes as out-of-distribution graphs and the remaining classes serve as normal graphs. One of the out-of-distribution classes is included in the training and validation sets, while the other is added to the testing set.

Table 5 presents the results of this set of experiments. Compared with other baseline models, our proposed RQGNN achieves significant improvements in both AUC and F1 scores in these new datasets, which demonstrates the effectiveness of our RQGNN in detecting anomalies beyond the application considered.

### A.5 ALGORITHM

---

**Algorithm 1:** RQL

    **Input:** $\{G_i\}$, where $i = 1, \cdots, n$
    **Output:** $\boldsymbol{H}^{G}_{RQ}$
**1** **for** $i = 1$ *to* $n$ **do**
**2**      $\tilde{\boldsymbol{X}} \leftarrow \text{MLP}(G_i.\boldsymbol{X})$;
**3**      $RQ \leftarrow diag(\frac{\tilde{\boldsymbol{X}}^T \boldsymbol{L} \tilde{\boldsymbol{X}}}{\tilde{\boldsymbol{X}}^T \tilde{\boldsymbol{X}}})$;
**4**      $\boldsymbol{h}^{G}_{RQ} \leftarrow \text{MLP}(RQ)$;
**5**      $(\boldsymbol{H}^{G}_{RQ})_i \leftarrow \boldsymbol{h}^{G}_{RQ}$;
**6** Return $\boldsymbol{H}^{G}_{RQ}$;

---

**Algorithm 2:** CWGNN with RQ-pooling

    **Input:** $\{G_i\}$, where $i = 1, \cdots, n$
    **Output:** $\boldsymbol{H}^{G}_{Att}$
**1** **for** $i = 1$ *to* $n$ **do**
**2**      $\tilde{\boldsymbol{X}} \leftarrow \text{MLP}(G_i.\boldsymbol{X})$;
**3**      $RQ \leftarrow diag(\frac{\tilde{\boldsymbol{X}}^T \boldsymbol{L} \tilde{\boldsymbol{X}}}{\tilde{\boldsymbol{X}}^T \tilde{\boldsymbol{X}}})$;
**4**      $\boldsymbol{h}^{G}_{Att} \leftarrow \boldsymbol{0}$;
**5**      **for** *node $j$ in* $G_i$ **do**
**6**          $\boldsymbol{h_j} \leftarrow \text{CONCAT}((f_1(\boldsymbol{L})\tilde{\boldsymbol{X}})_j, \cdots, (f_q(\boldsymbol{L})\tilde{\boldsymbol{X}})_j)$;
**7**          $a_j \leftarrow RQ \cdot \boldsymbol{h_j}$;
**8**          $\boldsymbol{h}^{G}_{Att} \leftarrow \boldsymbol{h}^{G}_{Att} + a_j \boldsymbol{h_j}$;
**9**      $\boldsymbol{h}^{G}_{Att} \leftarrow \sigma(\boldsymbol{h}^{G}_{Att})$;
**10**      $(\boldsymbol{H}^{G}_{Att})_i \leftarrow \boldsymbol{h}^{G}_{Att}$;
**11** Return $\boldsymbol{H}^{G}_{Att}$;

---

**Algorithm 3:** RQGNN

    **Input:** $\{G_i\}$, where $i = 1, \cdots, n$
    **Output:** $\boldsymbol{H}^{G}$
**1** $\boldsymbol{H}^{G}_{RQ} \leftarrow \text{RQL}(\{G_i\})$;
**2** $\boldsymbol{H}^{G}_{Att} \leftarrow \text{CWGNN with RQ-pooling}(\{G_i\})$;
**3** $\boldsymbol{H}^{G} \leftarrow \text{CONCAT}(\boldsymbol{H}^{G}_{RQ}, \boldsymbol{H}^{G}_{Att})$;
**4** Return $\boldsymbol{H}^{G}$;

---

As described in Section 3, RQGNN contains two components, RQL and Chebyshev Wavelet GNN with RQ-pooling. The RQL learns the explicit representation of the Rayleigh Quotient of graphs while the Chebyshev Wavelet GNN with RQ-pooling learns the implicit representation of graphs

guided by the Rayleigh Quotient. After getting the two representations, we combine them to obtain the final representation of graphs.

## A.6 PARAMETER ANALYSIS

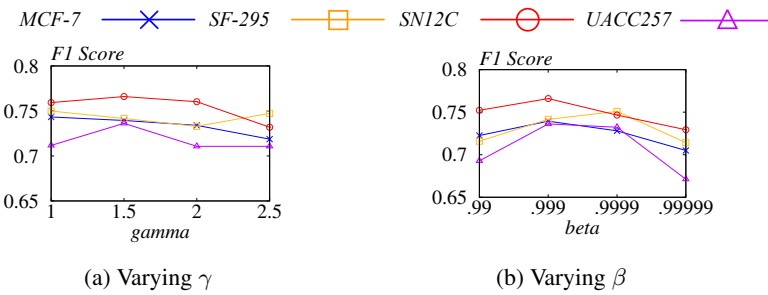

(a) Varying $\gamma$        (b) Varying $\beta$

Figure 4: Varying the $\gamma$ and $\beta$.

Other than the hyperparameters for the Chebyshev Wavelet GNN with RQ-pooling, we also investigate the impact of varying $\gamma$ and $\beta$ in the class-balanced focal loss on the performance of RQGNN. The results are shown in Figure 4. As we can see, when the $\gamma$ is set to 1.5, RQGNN achieves a relatively satisfactory performance on these four datasets. Meanwhile, RQGNN is relatively satisfactory when $\beta$ is set to 0.999. Hence, we choose 1.5 for $\gamma$ and 0.999 for $\beta$ in our experiments on all datasets.

## A.7 PERTURBATION ON SN12C

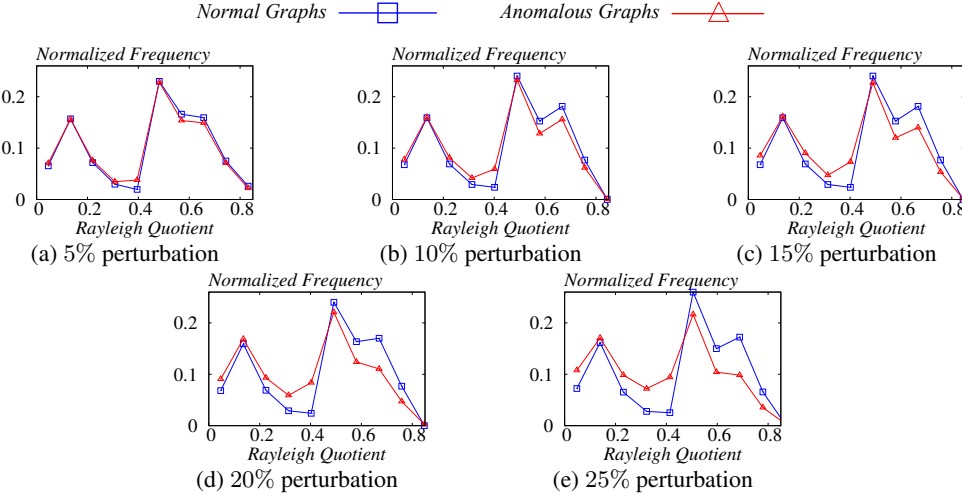

Figure 5: Normalized Rayleigh Quotient distribution on perturbated SN12C.

Recap from Section 3.1 that Theorem 1 and Theorem 2 show that the change of the Rayleigh Quotient can be bounded given a small perturbation on graph signal $x$ and graph Laplacian $L$. In this section, we further conduct experiments to explore how much perturbation is detectable by the proposed RQGNN. We use SN12C to create 5 new synthetic datasets. Specifically, we first randomly select 5% normal graphs to perform perturbations. For these graphs, each edge is changed with probability $p$, where $p = 0.05, 0.10, 0.15, 0.20, 0.25$. Then we use these perturbed samples as anomalous graphs and all the remaining 95% normal graphs in SN12C to construct these five new datasets. The corresponding normalized Rayleigh Quotient is shown in Figure 5.

As we can see from Figure 6, even with a perturbation probability as low as 0.05, RQGNN can still achieve a reasonable result. As the perturbation probability increases to 0.15, our RQGNN achieves an AUC score around 1 and an F1 score exceeding 0.9, which clearly demonstrates that RQGNN can accurately distinguish the two classes.

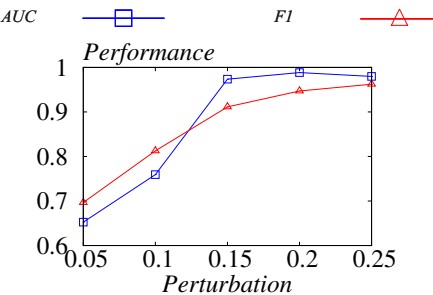

Figure 6: Performance of SN12C with different ratios of perturbation.

## A.8  DISTANCE RATIO BETWEEN DIFFERENT CLASSES

We calculate pairwise inter-distances and pairwise intra-distances for the normalized Rayleigh Quotient values of two classes. To keep consistency with the experiments shown in Figure 1, we still construct 10 equal-width bins. Subsequently, we report the results of inter-distances divided by intra-distances of the normalized Rayleigh Quotient for two classes, respectively. The detailed experimental results are presented in Table 6, where $d_i$ is the inter-distance between normalized values of Rayleigh Quotient of different classes, $d_a$ is the intra-distance between normalized values of Rayleigh Quotient values of anomalous graphs, and $d_n$ is the intra-distance between normalized values of Rayleigh Quotient values of normal graphs.

Table 6: Distance ratio between different classes of different datasets.

| Datasets | Ratio | $bin_0$ | $bin_1$ | $bin_2$ | $bin_3$ | $bin_4$ | $bin_5$ | $bin_6$ | $bin_7$ | $bin_8$ | $bin_9$ |
|---|---|---|---|---|---|---|---|---|---|---|---|
| MCF-7 | $d_i/d_a$ | 3.2630 | 2.7014 | 1.1659 | 5.4298 | 5.8771 | 3.5399 | 1.7634 | 0.8756 | 0.9013 | 17.6960 |
| | $d_i/d_n$ | 2.1631 | 5.2946 | 1.0262 | 4.2198 | 4.5321 | 7.5492 | 3.4830 | 1.8955 | 1.3387 | 7.8023 |
| MOLT-4 | $d_i/d_a$ | 3.4385 | 7.0977 | 0.4458 | 36.4909 | 7.1165 | 3.8262 | 2.2038 | 1.1997 | 0.2470 | 4.9966 |
| | $d_i/d_n$ | 2.1167 | 1.7348 | 3.8065 | 14.6213 | 41.3865 | 19.1898 | 5.1109 | 1.8908 | 0.5795 | 7.3194 |
| PC-3 | $d_i/d_a$ | 188.8639 | 4.8383 | 0.5602 | 4.4561 | 16.9550 | 4.5858 | 2.2441 | 0.7786 | 0.9506 | 7.2704 |
| | $d_i/d_n$ | 2.7716 | 8.9202 | 0.8078 | 4.9309 | 6.1323 | 11.7506 | 4.3151 | 2.0752 | 1.6180 | 6.4001 |
| SW-620 | $d_i/d_a$ | 10.0290 | 4.7585 | 2.6133 | 6.4650 | 5.5431 | 5.6535 | 1.8690 | 0.8894 | 0.6743 | 22.8407 |
| | $d_i/d_n$ | 2.4499 | 2.3840 | 50.2978 | 20.2884 | 299.8270 | 34.37006 | 6.0213 | 1.3747 | 1.3095 | 7.0025 |
| NCI-H23 | $d_i/d_a$ | 6.4151 | 7.6458 | 0.1815 | 9.6685 | 34.2458 | 6.8839 | 2.1601 | 0.3966 | 1.4374 | 38.5681 |
| | $d_i/d_n$ | 2.6216 | 2.7331 | 52.7906 | 15.1957 | 44.8589 | 33.3045 | 4.9554 | 0.7743 | 2.1844 | 6.3351 |
| OVCAR-8 | $d_i/d_a$ | 4.9954 | 19.6767 | 1.2632 | 8.7049 | 19.3535 | 3.9187 | 2.1573 | 0.8668 | 0.8162 | 113.6427 |
| | $d_i/d_n$ | 2.6150 | 2.9819 | 10.5372 | 14.4965 | 103.3491 | 35.3353 | 4.6184 | 1.7530 | 1.6612 | 6.5138 |
| P388 | $d_i/d_a$ | 8.9957 | 8.3098 | 0.3023 | 0.4619 | 1.7153 | 0.7943 | 1.3958 | 11.0056 | 0.2758 | 1.1250 |
| | $d_i/d_n$ | 2.0826 | 6.9907 | 1.1292 | 2.1624 | 4.6504 | 0.8967 | 10.5379 | 3.2803 | 1.3869 | 8.6336 |
| SF-295 | $d_i/d_a$ | 11.2356 | 5.2296 | 0.2031 | 81.6818 | 3.7652 | 3.7081 | 2.4469 | 0.6470 | 0.9760 | 17.3201 |
| | $d_i/d_n$ | 2.5468 | 3.1328 | 3.2846 | 12.9510 | 67.4922 | 31.3818 | 5.7962 | 1.2385 | 1.9829 | 7.4618 |
| SN12C | $d_i/d_a$ | 5.2940 | 8.8330 | 0.5253 | 8.9315 | 11.8474 | 6.2311 | 1.9104 | 0.6942 | 0.7776 | 114.3876 |
| | $d_i/d_n$ | 3.0953 | 2.4427 | 16.7566 | 17.5596 | 114.7632 | 30.4557 | 5.3002 | 1.2671 | 1.4012 | 5.6286 |
| UACC257 | $d_i/d_a$ | 4.8530 | 33.1566 | 0.2014 | 5.3367 | 8.0657 | 8.8780 | 2.0177 | 0.9534 | 1.3980 | 38.5248 |
| | $d_i/d_n$ | 2.5624 | 2.4816 | 2.1689 | 25.3848 | 225.1055 | 52.8033 | 4.7289 | 1.8897 | 2.1000 | 6.1323 |

Additionally, we present the distance ratios between different classes of different perturbations (recap from Section A.7) on SN12C in Table 7.

Table 7: Distance ratio between different classes of different perturbations

| Datasets | Ratio | $bin_0$ | $bin_1$ | $bin_2$ | $bin_3$ | $bin_4$ | $bin_5$ | $bin_6$ | $bin_7$ | $bin_8$ | $bin_9$ |
|---|---|---|---|---|---|---|---|---|---|---|---|
| 5% perturbation | $d_i/d_a$ | 7.3235 | 0.0882 | 0.8025 | 2.1880 | 13.4657 | 0.1622 | 1.5447 | 2.9775 | 0.7854 | 1.9094 |
| | $d_i/d_n$ | 0.6507 | 2.6996 | 2.8406 | 4.8837 | 9.9019 | 1.1670 | 1.3218 | 1.1467 | 0.6372 | 8.0939 |
| 10% perturbation | $d_i/d_a$ | 23.2792 | 0.1572 | 2.2364 | 37.7247 | 19.0277 | 1.0230 | 2.7819 | 6.6939 | 9.5345 | 0.4399 |
| | $d_i/d_n$ | 1.2710 | 2.1388 | 10.3101 | 16.4515 | 8.5852 | 1.0900 | 2.5552 | 2.2331 | 4.0627 | 9.0571 |
| 15% perturbation | $d_i/d_a$ | 24.8715 | 0.3862 | 4.8172 | 27.4571 | 16.0609 | 1.8245 | 5.2849 | 35.0558 | 300.1928 | 0.4652 |
| | $d_i/d_n$ | 2.2010 | 5.0905 | 18.0752 | 23.4784 | 11.9045 | 1.8214 | 3.4373 | 3.6316 | 6.2911 | 15.2549 |
| 20% perturbation | $d_i/d_a$ | 16.4177 | 0.8982 | 3.1912 | 13.8354 | 25.3389 | 3.3763 | 33.8295 | 12.6984 | 26.2688 | 0.4820 |
| | $d_i/d_n$ | 2.7813 | 23.0458 | 18.3562 | 43.5881 | 14.4079 | 2.6532 | 4.8426 | 5.8488 | 8.0527 | 27.8505 |
| 25% perturbation | $d_i/d_a$ | 20.5731 | 1.0082 | 6.9559 | 29.4320 | 19.3459 | 7.7016 | 16.7429 | 23.0523 | 14.1050 | 0.4837 |
| | $d_i/d_n$ | 4.2974 | 9.3829 | 37.7505 | 68.5836 | 12.8552 | 5.1918 | 8.7394 | 7.9608 | 7.9395 | 31.4415 |

As we can see, there is no consistent pattern in the numerical data presented in the two tables. Nevertheless, RQGNN successfully learns the Rayleigh Quotient of the graphs and effectively distinguishes between the two classes. This demonstrates that a consistent pattern in the ratio is not necessary for accurate classification. The key to differentiating between the classes is capturing and leveraging such gaps on certain dimensions, rather than relying on consistent patterns in the data.

### A.9 RAYLEIGH QUOTIENT DISTRIBUTION ON OTHER DATASETS

To further demonstrate the Rayleigh Quotient can be applied to any graph-level anomaly detection datasets, we calculate the normalized Rayleigh Quotient distribution of all the other 9 datasets, which are shown in Figures 7-15. As we can observe, for each dataset, regardless of the variations in the sample size, the normalized Rayleigh Quotient distribution of each class exhibits a consistent pattern across different sample sets. Besides, the normalized Rayleigh Quotient distribution of anomalous graphs and that of normal ones are statistically distinct from each other on all datasets. These results again demonstrate that the Rayleigh Quotient can reveal the underlying differences between normal and anomalous graphs.

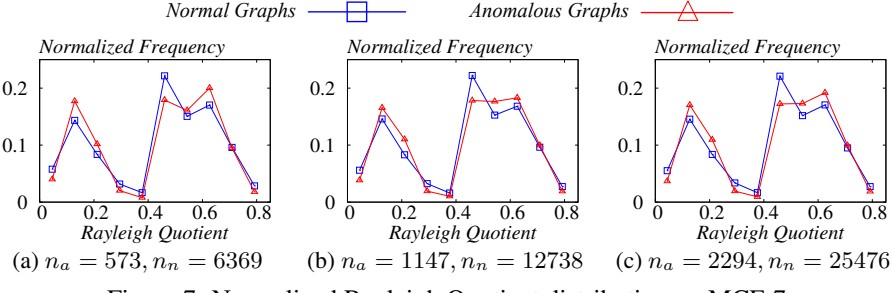

Figure 7: Normalized Rayleigh Quotient distribution on MCF-7.

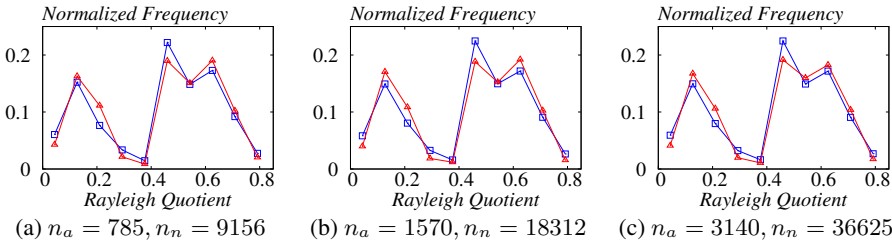

Figure 8: Normalized Rayleigh Quotient distribution on MOLT-4.

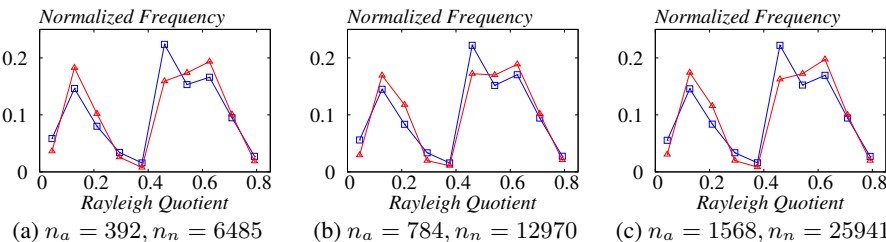

Figure 9: Normalized Rayleigh Quotient distribution on PC-3.

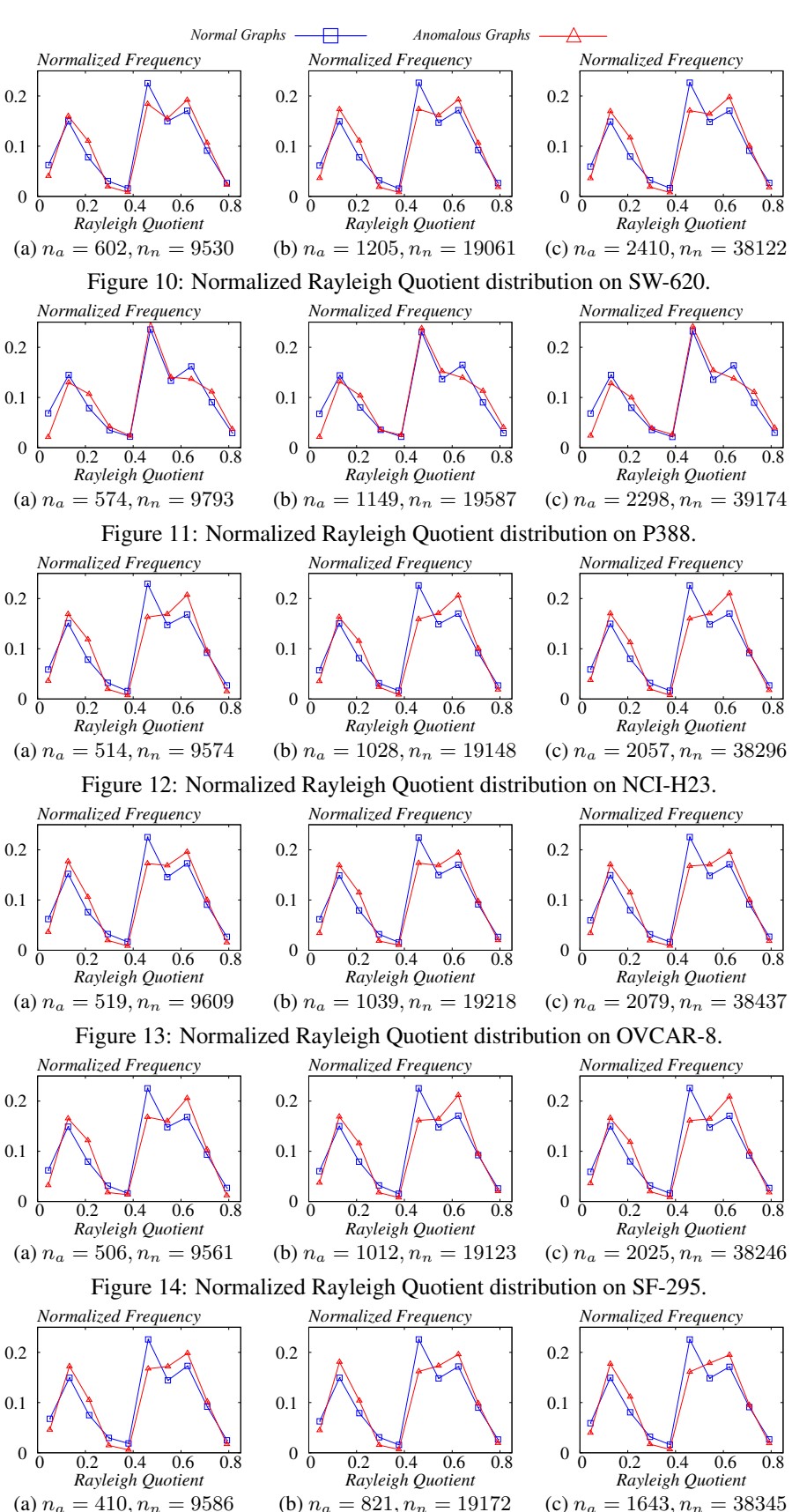

Figure 10: Normalized Rayleigh Quotient distribution on SW-620.

Figure 11: Normalized Rayleigh Quotient distribution on P388.

Figure 12: Normalized Rayleigh Quotient distribution on NCI-H23.

Figure 13: Normalized Rayleigh Quotient distribution on OVCAR-8.

Figure 14: Normalized Rayleigh Quotient distribution on SF-295.

Figure 15: Normalized Rayleigh Quotient distribution on UACC257.

