# OpenReview forum: "Rayleigh Quotient Graph Neural Networks for Graph-level Anomaly Detection"
_ICLR.cc/2024/Conference — ICLR 2024 poster_

### Official Review · Reviewer_X1rz · 2023-10-20

**Soundness:** 2 fair
**Presentation:** 2 fair
**Contribution:** 2 fair
**Rating:** 6
**Confidence:** 4

**Summary:**

This paper proposes the first spectral GNN-based graph-level anomaly detection method named RQGNN. Considering the disparity in the accumulated spectral energy between the anomalies and normal graphs, RQGNN leverages the Rayleigh Quotient learning component to capture the accumulated spectral energy of graphs. In addition, RQGNN also proposes the Chebyshev Wavelet GNN to represent nodes. RQ-pooling that regards the Rayleigh Quotient coefficient as the node importance score will be employed to obtain graph representation. The final graph representation is the concatenation of the Rayleigh Quotient learning component and the Chebyshev Wavelet GNN. Finally, class-balanced focal loss is introduced to optimize RQGNN and obtain graphs’ binary labels.

**Strengths:**

1.	The topic of graph-level anomaly detection focused by this paper is an interesting but under-explored research area.
2.	This paper is the first to consider solving the problem of graph-level anomaly detection from the perspective of spectral graph. Before this, leveraging graph wavelet to identify anomalous nodes within a single graph has achieved empirical successes [1].
3.	This paper proposes a sufficient survey on graph anomaly detection.

[1] Tang et al. Rethinking Graph Neural Networks for Anomaly Detection. ICML 2022.

**Weaknesses:**

1.	The insight of this paper seems to be on shaky ground. Figures 1 and 4-12 do not clearly show the statistical difference between anomalous graphs and normal graphs with respect to Rayleigh Quotient. Detailed text description about the ''significant disparity" between two classes is necessary but not found on the current version. (This is the main reason why I currently tend to reject this paper.)
2.	I recommend briefly introducing Rayleigh Quotient in the introduction or preliminaries section.
3.	In page 4,  authors wrote  "If the graph Laplacian $\mathbf L$ and graph signal $\mathbf x$ of two graphs are close, then their Rayleigh Quotients will be close to each other and these two graphs will highly likely belong to the same class." I tend to think this statement is correct, but it seems not applicable for anomalies. Anything that is different from normal can be regarded as an anomaly, but we actually cannot get training data that can represent the full picture of anomalies. In addition, two anomalous graphs can also be very different. This paper uses two-class datasets for experiments, but what will be the result if we regard the third class that has never appeared in the training set as anomalies to test the proposed model?
4.	In page 5, authors wrote "However, as analyzed in Section 3.1, to capture the spectral properties of anomalous graphs, it is necessary to consider the spectral energy with respect to each eigenvalue." But, the reason for using graph wavelet convolution is still unclear.

**Questions:**

1. How about the graph-level anomaly detection performance of ChebyNet, BernNet, GMT, Gmixup, and TVGNN if the loss function is replaced by the class-balanced focal loss?
2. What is the ratio between the normal and anomalous graphs utilized during model training?

---

> ### Author Response · Authors · 2023-11-19
> **Author Response (1/3)**
>
> We appreciate your comprehensive and constructive review. Your crucial comments on the statistical differences between anomalous graphs and normal graphs are exceedingly helpful for us to improve our manuscript. Our point-to-point responses to your comments are given below.
>
> ---
>
> **W1**: The insight of this paper seems to be on shaky ground. Figures 1 and 4-12 do not clearly show the statistical difference between anomalous graphs and normal graphs with respect to Rayleigh Quotient. Detailed text description about the ''significant disparity" between two classes is necessary but not found on the current version. (This is the main reason why I currently tend to reject this paper.)
>
> **RW1**: Thanks for your constructive suggestions. Your feedback will greatly contribute to improving the quality of our manuscript. Our detailed explanation is as follows.
>
> In the field of graph-level anomaly detection, researchers mainly rely on the chemical compound datasets available from TUDataset [2]. To the best of our knowledge, these datasets are the only datasets directly applicable to the graph-level anomaly detection task. Each dataset consists of several chemical compounds (graphs) whose labels are decided by whether they are inactive or active to a tumor. Despite being categorized into two classes, these chemical compounds still share many similar characteristics, so it is natural that the Rayleigh Quotient of these two classes presents similarity in some dimensions, which reflects the reviewer’s concern.
>
> However, it is important to note that the values of Rayleigh Quotient on anomalous graphs and normal graphs can be extremely distinct from each other in some specific dimensions, showing the key differences between the compounds in the two classes. Our proposed model is capable of capturing these differences and hence achieves better performance as shown in the experiment. To better show the “significant disparity” of the Rayleigh Quotient between anomalous graphs and normal graphs, we further add the following experiment.
>
> In particular, we calculate the intra-distance of the normalized Rayleigh Quotient on normal graphs and anomalous graphs, respectively. Besides, we calculate the inter-distance of the normalized Rayleigh Quotient between normal graphs and anomalous graphs.
>
> Due to the time limit, we present the result on the SN12C dataset. We first calculate the pairwise inter-distances and pairwise intra-distances for the normalized Rayleigh Quotient of two classes. To keep consistency with the experiments described in Figure 1 (refer to Section 1),  we still construct 10 equal-width bins. Subsequently, we report the results of inter-distances divided by intra-distances of the normalized Rayleigh Quotient for the two classes, respectively. The detailed experimental result is presented below, where $d_i / d_a = \frac{\text{inter-distance between normalized RQ of different classes}}{\text{intra-distance between normalized RQ of anomalous graphs}}$ and $d_i / d_n=\frac{\text{inter-distance between normalized RQ of different classes}}{\text{intra-distance between normalized RQ of normal graphs}}$.
>
> |          Ratio          |    bin0    |   bin1   |    bin2    |   bin3    |   bin4    |   bin5    |   bin6    |   bin7    |    bin8    |   bin9   |
> | :---------------------: | :------: | :-----: | :------: | :-----: | :-----: | :-----: | :-----: | :-----: | :------: | :-----: |
> | $d_i / d_a$  | 5.2940 | 8.8330 | 0.5253 | 8.9315 | 11.8474 | 6.2311 | 1.9104 | 0.6942 | 0.7776 | 114.3876 |
> | $d_i / d_n$ | 3.0953  | 2.4427  | 16.7566  | 17.5596 | 114.7632  | 30.4557 | 5.3002  | 1.2671  |  1.4012  | 5.6286  |
>
> As shown in the above table, the result clearly shows a significant disparity between the inter-distance and intra-distance values. Notice that in certain cases, the values of inter-distances can be 10 times or even 100 times larger than that of intra-distances, and the larger the ratios are, the more distinguishable these two classes are. Moreover, it is apparent that normal and anomalous graphs exhibit different patterns in many cases. which clearly demonstrates the “significant disparity” between the two classes.
>
> [2] https://chrsmrrs.github.io/datasets/docs/datasets/
>
> ---
>
> **W2**: I recommend briefly introducing Rayleigh Quotient in the introduction or preliminaries section.
>
> **RW2**: Thanks for your constructive suggestions. We will incorporate the relevant content into our revised manuscript.
>
> ---

---

> ### Author Response · Authors · 2023-11-19
> **Author Response (2/3)**
>
> **W3**: In page 4, authors wrote "If the graph Laplacian and graph signal of two graphs are close, then their Rayleigh Quotients will be close to each other and these two graphs will highly likely belong to the same class." I tend to think this statement is correct, but it seems not applicable for anomalies. Anything that is different from normal can be regarded as an anomaly, but we actually cannot get training data that can represent the full picture of anomalies. In addition, two anomalous graphs can also be very different. This paper uses two-class datasets for experiments, but what will be the result if we regard the third class that has never appeared in the training set as anomalies to test the proposed model?
>
> **RW3**: Thanks for raising this point. The problem you mentioned can be categorized as an out-of-distribution detection task rather than an anomaly detection task. We acknowledge that this problem is out of the scope of this manuscript since there are notable distinctions between anomaly detection and out-of-distribution detection.
>
> Typically, in out-of-distribution datasets, we only have one class called in-distribution data, so out-of-distribution detection focuses on determining whether a new sample belongs to the in-distribution data. However, anomaly datasets usually contain data with two distinct characteristics. Therefore, anomaly detection aims to determine that a new sample belongs to one of the two classes.
>
> For example, anomaly detection involves identifying all the tigers from a dataset that contains both cats and tigers, whereas out-of-distribution detection focuses on determining whether a new cat sample, such as a Birman cat, can be correctly classified as a cat given training dataset that exclusively contains Ragdoll cats.
>
> In our manuscript, we follow the settings of iGAD and further include additional related baselines to conduct experiments. To the best of our knowledge, the datasets we employ are the only ones available in the field of graph-level anomaly detection.
>
> Despite our primary focus not being on the out-of-distribution task, we still conduct additional experiments to evaluate the performance of our proposed model. We use a modified COLORS-3 [1] dataset from TUDataset [2], which is an 11-class graph classification dataset. We randomly select two classes as out-of-distribution graphs and the remaining classes serve as normal graphs. One of the out-of-distribution classes is included in the training and validation sets, while the other is added to the testing set.
>
> The results of this set of experiments are as follows:
>
> | Metrics |   RQGNN   |  iGAD  | HimNet |  GMT   | TVGNN  | BernNet | ChebyNet |
> | :-----: | :--------: | :----: | :----: | :----: | :----: | :-----: | :------: |
> |   AUC   | **0.9421** | 0.7630 | 0.4522 | 0.5000 |  0.5000   | 0.6486  |  0.6500  |
> |   F1    | **0.8423** | 0.6674 | 0.4707 | 0.3710 | 0.3710 | 0.3710  |  0.3710  |
>
> Compared with other baseline models, our proposed RQGNN achieves significant improvements in both AUC and F1 scores although RQGNN does not focus on out-of-distribution detection.
>
> We will include these results in our revised manuscript.
>
> [1] Boris Knyazev, Graham W. Taylor, Mohamed R. Amer. Understanding attention and generalization in graph neural networks. NeurIPS, pages 4204-4214, 2019.
>
> [2] https://chrsmrrs.github.io/datasets/docs/datasets/
>
> ---
>
> **W4**: In page 5, authors wrote "However, as analyzed in Section 3.1, to capture the spectral properties of anomalous graphs, it is necessary to consider the spectral energy with respect to each eigenvalue." But, the reason for using graph wavelet convolution is still unclear.
>
> **RW4**: Graph wavelet convolution has been demonstrated to possess a strong capability to learn representations in the spectral domain from previous research, making it a natural choice as our backbone model. We also compared our RQGNN with other spectral GNNs, such as BernNet and ChebyNet, but as shown in our experiments in Section 4, even the state-of-the-art spectral GNN models can not handle complex graph-level anomaly detection tasks.
>
> In addition, as proven in Section 3, we need to consider spectral energy in terms of each eigenvalue. According to BernNet [8], each graph filter can be viewed as a band filter, and each band filter only allows certain spectral energy to pass, so it is necessary to adopt a model with multiple graph filters. The graph wavelet convolution model can be considered as a combination of different graph filters, enabling us to consider the spectral energy associated with each eigenvalue. By employing different band filters, we can focus on the spectral energy of different eigenvalues. Therefore, leveraging graph wavelet convolution provides advantages compared to using single graph filters.
>
> [8] Bernnet: Learning arbitrary graph spectral filters via bernstein approximation. In NeurIPS, pp. 14239–14251, 2021.
>
> ---

---

> ### Author Response · Authors · 2023-11-19
> **Author Response (3/3)**
>
> **Q1**: How about the graph-level anomaly detection performance of ChebyNet, BernNet, GMT, Gmixup, and TVGNN if the loss function is replaced by the class-balanced focal loss?
>
> **RQ1**: We have conducted experiments for GMT, Gmixup, and TVGNN. Please refer to Table 3 in the Appendix in the original manuscript for detailed experimental results.
>
> The following table reports the results of ChebyNet and BernNet, where BernNet-CB and ChebyNet-CB denote the corresponding model with the class-balanced focal loss function, respectively.
>
> | Datasets | Metrics | BernNet | BernNet-CB | ChebyNet | ChebyNet-CB | RQGNN  |
> | :------: | :-----: | :-----: | :-------------: | :------: | :--------------: | :----: |
> |  MCF-7   |   AUC   | 0.6172  |     0.6215      |  0.6612  |      0.6617      | 0.8354 |
> |          |   F1    | 0.4784  |     0.4784      |  0.4780  |      0.4804      | 0.7394 |
> |  MOLT-4  |   AUC   | 0.6144  |     0.6068      |  0.6647  |      0.6650      | 0.8316 |
> |          |   F1    | 0.4794  |     0.4794      |  0.4854  |      0.4875      | 0.7240 |
> |   PC-3   |   AUC   | 0.6094  |     0.6040      |  0.6051  |      0.6065      | 0.8782 |
> |          |   F1    | 0.4853  |     0.4853      |  0.4853  |      0.4853      | 0.7184 |
> |  SW-620  |   AUC   | 0.6072  |     0.6081      |  0.6759  |      0.6823      | 0.8550 |
> |          |   F1    | 0.4847  |     0.4847      |  0.4898  |      0.4947      | 0.7335 |
> | NCI-H23  |   AUC   | 0.6114  |     0.6289      |  0.6728  |      0.6734      | 0.8680 |
> |          |   F1    | 0.4869  |     0.4869      |  0.4930  |      0.5203      | 0.7214 |
> | OVCAR-8  |   AUC   | 0.5850  |     0.5803      |  0.6303  |      0.6294      | 0.8799 |
> |          |   F1    | 0.4868  |     0.4868      |  0.4900  |      0.4992      | 0.7215 |
> |   P388   |   AUC   | 0.6707  |     0.6694      |  0.7266  |      0.7324      | 0.9023 |
> |          |   F1    | 0.5001  |     0.5002      |  0.5635  |      0.5656      | 0.7963 |
> |  SF-295  |   AUC   | 0.6353  |     0.6302      |  0.6650  |      0.6670      | 0.8825 |
> |          |   F1    | 0.4871  |     0.4871      |  0.4871  |      0.4871      | 0.7416 |
> |  SN12C   |   AUC   | 0.6014  |     0.5992      |  0.6598  |      0.6634      | 0.8861 |
> |          |   F1    | 0.4874  |     0.4874      |  0.4972  |      0.4970      | 0.7660 |
> | UACC257  |   AUC   | 0.6115  |     0.6130      |  0.6584  |      0.6645      | 0.8724 |
> |          |   F1    | 0.4895  |     0.4895      |  0.4894  |      0.4933      | 0.7362 |
>
> As we can see, the class-balanced focal loss does not yield benefits for ChebyNet and BernNet. This further demonstrates that even though tackling the imbalanced problem of datasets, these spectral GNNs fail to capture the properties of anomalous graphs.
>
> We will include these results in our revised manuscript.
>
> ---
> **Q2**: What is the ratio between the normal and anomalous graphs utilized during model training?
>
> **RQ2**: Please refer to Table 1 in the original manuscript for the statistical information of all datasets, which includes the number of normal and anomalous graphs. The anomalous ratio for each dataset is around 5%. In addition, in Section 4, we have stated that “we randomly divide each dataset into training/validation/test sets with 70%/15%/15%, respectively. During the sampling process, we ensure that each set maintains a consistent ratio between normal and anomalous graphs.” Therefore, the ratio between normal and anomalous graphs utilized during model training for each dataset remains consistent with the original dataset.
>
> ---
>
> We sincerely appreciate your time, and we are glad to answer any additional questions you may have.

---

> > ### Comment · Reviewer_X1rz · 2023-11-20
> > **Thanks for the reply.**
> >
> > The data presented in the manuscript (figure 1, figures 4-12) indicates that this contrast in Rayleigh Quotient between normal and anomalous graphs  is not statistically significant. The current version of these figures does not help this article be published. Despite the authors show that the SN12C dataset exhibits a substantial difference in Rayleigh Quotient between normal and anomalous graphs in bin4 and bin9, I am apprehensive about the consistency of this pattern across other datasets. As this forms the foundational premise of the paper, I am particularly concerned about addressing this issue.

---

> > > ### Author Response · Authors · 2023-11-20
> > > **Author Response (1/1)**
> > >
> > > ---
> > >
> > > **C1**: The data presented in the manuscript (figure 1, figures 4-12) indicates that this contrast in Rayleigh Quotient between normal and anomalous graphs is not statistically significant. The current version of these figures does not help this article be published. Despite the authors show that the SN12C dataset exhibits a substantial difference in Rayleigh Quotient between normal and anomalous graphs in bin4 and bin9, I am apprehensive about the consistency of this pattern across other datasets. As this forms the foundational premise of the paper, I am particularly concerned about addressing this issue.
> > >
> > > **RC1**: Thanks for your constructive suggestion. To further address your concern, we conduct the same experiments on the other 9 datasets. We present the results here:
> > >
> > > | Datasets |   Ratio   |   bin0   |  bin1   |  bin2   |  bin3   |   bin4   |  bin5   |  bin6   |  bin7   |  bin8  |   bin9   |
> > > | :------: | :-------: | :------: | :-----: | :-----: | :-----: | :------: | :-----: | :-----: | :-----: | :----: | :------: |
> > > |  MCF-7   | $d_i/d_a$ |  3.2630  | 2.7014  | 1.1659  | 5.4298  |  5.8771  | 3.5399  | 1.7634  | 0.8756  | 0.9013 | 17.6960  |
> > > |          | $d_i/d_n$ |  2.1631  | 5.2946  | 1.0262  | 4.2198  |  4.5321  | 7.5492  | 3.4830  | 1.8955  | 1.3387 |  7.8023  |
> > > |  MOLT-4  | $d_i/d_a$ |  3.4385  | 7.0977  | 0.4458  | 36.4909 |  7.1165  | 3.8262  | 2.2038  | 1.1997  | 0.2470 |  4.9966  |
> > > |          | $d_i/d_n$ |  2.1167  | 1.7348  | 3.8065  | 14.6213 | 41.3865  | 19.1898 | 5.1109  | 1.8908  | 0.5795 |  7.3194  |
> > > |   PC-3   | $d_i/d_a$ | 188.8639 | 4.8383  | 0.5602  | 4.4561  | 16.9550  | 4.5858  | 2.2441  | 0.7786  | 0.9506 |  7.2704  |
> > > |          | $d_i/d_n$ |  2.7716  | 8.9202  | 0.8078  | 4.9309  |  6.1323  | 11.7506 | 4.3151  | 2.0752  | 1.6180 |  6.4001  |
> > > |  SW-620  | $d_i/d_a$ | 10.0290  | 4.7585  | 2.6133  | 6.4650  |  5.5431  | 5.6535  | 1.8690  | 0.8894  | 0.6743 | 22.8407  |
> > > |          | $d_i/d_n$ |  2.4499  | 2.3840  | 50.2978 | 20.2884 | 299.8270 | 34.3706 | 6.0213  | 1.3747  | 1.3095 |  7.0025  |
> > > | NCI-H23  | $d_i/d_a$ |  6.4151  | 7.6458  | 0.1815  | 9.6685  | 34.2458  | 6.8839  | 2.1601  | 0.3966  | 1.4374 | 38.5681  |
> > > |          | $d_i/d_n$ |  2.6216  | 2.7331  | 52.7906 | 15.1957 | 44.8589  | 33.3045 | 4.9554  | 0.7743  | 2.1844 |  6.3351  |
> > > | OVCAR-8  | $d_i/d_a$ |  4.9954  | 19.6767 | 1.2632  | 8.7049  | 19.3535  | 3.9187  | 2.1573  | 0.8668  | 0.8162 | 113.6427 |
> > > |          | $d_i/d_n$ |  2.6150  | 2.9819  | 10.5372 | 14.4965 | 103.3491 | 35.3353 | 4.6184  | 1.7530  | 1.6612 |  6.5138  |
> > > |   P388   | $d_i/d_a$ |  8.9957  | 8.3098  | 0.3023  | 0.4619  |  1.7153  | 0.7943  | 1.3958  | 11.0056 | 0.2758 |  1.1250  |
> > > |          | $d_i/d_n$ |  2.0826  | 6.9907  | 1.1292  | 2.1624  |  4.6504  | 0.8967  | 10.5379 | 3.2803  | 1.3869 |  8.6336  |
> > > |  SF-295  | $d_i/d_a$ | 11.2356  | 5.2296  | 0.2031  | 81.6818 |  3.7652  | 3.7081  | 2.4469  | 0.6470  | 0.9760 | 17.3201  |
> > > |          | $d_i/d_n$ |  2.5468  | 3.1328  | 3.2846  | 12.9510 | 67.4922  | 31.3818 | 5.7962  | 1.2385  | 1.9829 |  7.4618  |
> > > |  SN12C   | $d_i/d_a$ |  5.2940  | 8.8330  | 0.5253  | 8.9315  | 11.8474  | 6.2311  | 1.9104  | 0.6942  | 0.7776 | 114.3876 |
> > > |          | $d_i/d_n$ |  3.0953  | 2.4427  | 16.7566 | 17.5596 | 114.7632 | 30.4557 | 5.3002  | 1.2671  | 1.4012 |  5.6286  |
> > > | UACC257  | $d_i/d_a$ |  4.8530  | 33.1566 | 0.2014  | 5.3367  |  8.0657  | 8.8780  | 2.0177  | 0.9534  | 1.3980 | 38.5248  |
> > > |          | $d_i/d_n$ |  2.5624  | 2.4816  | 2.1689  | 25.3848 | 225.1055 | 52.8033 | 4.7289  | 1.8897  | 2.1000 |  6.1323  |
> > >
> > > As we can see from the table, the results of all the datasets present a large gap between the inter-distance and the intra-distance on certain dimensions. On all datasets, the largest disparity of $d_i / d_a$ or $d_i / d_n$ is at least 10 and even up to around 300. For instance, the result of SW-620 in bin4 of $d_i / d_n$ is 299.8270. These results demonstrate the “significant disparity” between the normalized Rayleion Quotient of two classes on all the tested datasets. These results together with our previous result in the original manuscript support the foundational premise of the paper.

---

> > > > ### Comment · Reviewer_X1rz · 2023-11-20
> > > >
> > > > Thank you for the author's response. Based on the information provided in this table, my current interpretation of the article is as follows. The Rayleigh Quotient serves as an auxiliary feature, employed by the author to assist Chebyshev Wavelet GNN in conducting graph-level anomaly detection, leveraging the subtle distinctions between anomalous and normal graphs. However, I did not discern clear patterns in the numerical data presented in this table (for instance, the Rayleigh Quotient of normal graphs consistently appears higher in certain bins than that of anomalous graphs). Therefore, I am concerned that the Rayleigh Quotient may not be specifically tailored for graph-level anomaly detection. Given this consideration, I am currently inclined to adjust my score to 5 as an expression of gratitude for the author's patient response.

---

> > > > > ### Author Response · Authors · 2023-11-21
> > > > > **Author Response (1/2)**
> > > > >
> > > > > ---
> > > > >
> > > > > **C2**: Based on the information provided in this table, my current interpretation of the article is as follows. The Rayleigh Quotient serves as an auxiliary feature, employed by the author to assist Chebyshev Wavelet GNN in conducting graph-level anomaly detection, leveraging the subtle distinctions between anomalous and normal graphs. However, I did not discern clear patterns in the numerical data presented in this table (for instance, the Rayleigh Quotient of normal graphs consistently appears higher in certain bins than that of anomalous graphs). Therefore, I am concerned that the Rayleigh Quotient may not be specifically tailored for graph-level anomaly detection.
> > > > >
> > > > > **RC2**: Thanks for your timely response. We will further make the following clarification to address the reviewer’s concern.
> > > > >
> > > > > Firstly, it is natural that different datasets have different patterns in terms of inter-distances and intra-distances of the normalized Rayleigh Quotient values. The main point is whether the inter-distance is way larger than the intra-distance on certain dimensions. If there is a gap on certain dimensions, then this means in these dimensions, our RQGNN can capture the difference so that we can distinguish two classes. Hence, there is no need for all the datasets to share the same pattern in terms of the ratio of inter-distance and intra-distance.
> > > > >
> > > > > Secondly, as shown in Section 3, the RQL will explicitly learn the Rayleigh Quotient of each data while the Chebyshev Wavelet GNN with RQ-pooling will implicitly learn the graph representation guided by the Rayleigh Quotient. For RQL, the difference on certain dimensions between normal and anomalous graphs will be explicitly captured due to the large ratio of inter-distance and intra-distance on these dimensions. For Chebyshev Wavelet GNN with RQ-pooling, it explores the spectral space of the graphs and learns the differences between normal and anomalous graphs guided by the Rayleigh Quotient. Since there is a huge gap between the two classes on certain dimensions in terms of inter-distance and intra-distance, no matter what the dimensions are, they will help Chebyshev Wavelet GNN with RQ-pooling encode the spectral information effectively so that our model can distinguish the two classes.
> > > > >
> > > > > We further conduct experiments to explore how much perturbation is detectable by RQGNN. In this set of experiments, we use SN12C to create 5 new synthetic datasets. Specifically, we first randomly select 5% normal graphs to perform perturbations. For these graphs, each edge is changed with probability $p$, where $p=0.05, 0.10, 0.15, 0.20,$ or $0.25$. Then we use these perturbed samples as anomalous graphs and all the remaining 95% normal graphs in SN12C to construct these five new datasets. The results are shown in the following table:
> > > > >
> > > > > | Metrics |   0.05  |  0.10 | 0.15 |  0.20  | 0.25 |
> > > > > | :-----: | :--------: | :----: | :----: | :----: | :----: |
> > > > > |   AUC   | 0.6525 | 0.7592 | 0.9736 | 0.9884 |  0.9798   |
> > > > > |   F1    | 0.6966 | 0.8126 | 0.9113 | 0.9471 | 0.9623 |
> > > > >
> > > > > As we can observe, even with a perturbation probability as low as 0.05, RQGNN can still achieve a reasonable result. As the probability of perturbation increases to 0.15, our RQGNN achieves an AUC score around 1 and an F1 score exceeding 0.9, which clearly demonstrates that RQGNN can accurately distinguish the two classes.
> > > > >
> > > > > We illustrate the normalized Rayleigh Quotient of these five new datasets in our revised manuscript to show the disparity between the normalized Rayleigh Quotient distribution of the two classes.

---

> > > > > ### Author Response · Authors · 2023-11-21
> > > > > **Author Response (2/2)**
> > > > >
> > > > > Additionally, we present the results of inter-distances divided by the intra-distances of these five datasets in the following table:
> > > > >
> > > > > | Datasets |   Ratio   |   bin0   |  bin1   |  bin2   |  bin3   |   bin4   |  bin5   |  bin6   |  bin7   |  bin8  |   bin9   |
> > > > > | :------: | :-------: | :------: | :-----: | :-----: | :-----: | :------: | :-----: | :-----: | :-----: | :----: | :------: |
> > > > > |  5% perturbation   | $d_i/d_a$ |  7.3235  | 0.0882  | 0.8025  | 2.1880  |  13.4657  | 0.1622  | 1.5447  | 2.9775  | 0.7854 | 1.9094  |
> > > > > |          | $d_i/d_n$ |  0.6507  | 2.6996  | 2.8406  | 4.8837  | 9.9019  |0.1670  | 1.3218  | 1.1467  | 0.6372 |  8.0939  |
> > > > > |  10% perturbation  | $d_i/d_a$ |  23.2792  | 0.1572  | 2.2364  | 37.7247 |  19.0277  | 1.0230  | 2.7819  | 6.6939  | 9.5345 |  0.4399  |
> > > > > |          | $d_i/d_n$ |  1.2710  | 2.1388 | 10.3101  | 16.4515 | 8.5852  | 1.0900 | 2.5552  |2.2331  | 4.0627 |  9.0571  |
> > > > > |   15% perturbation   | $d_i/d_a$ | 24.8715 | 0.3862  | 4.8172  |27.4571  | 16.0609 | 1.8245  | 5.2849  | 35.0558  | 300.1928 |  0.4652 |
> > > > > |          | $d_i/d_n$ |  2.2010  | 5.0905  | 18.0752  | 23.4784 |  11.9045 | 1.8214 | 3.4373  |3.6316 | 6.2911 |  15.2549  |
> > > > > |  20% perturbation  | $d_i/d_a$ | 16.4177  | 0.8982  | 3.1912  | 13.8354 |  25.3389  |3.3763  | 33.8295  | 12.6984 |26.2688 | 0.4820 |
> > > > > |          | $d_i/d_n$ |  2.7813  | 23.0458  | 18.3562 | 43.5881 | 14.4079 | 2.6532 | 4.8426  | 5.8488  | 8.0527 |  27.8505 |
> > > > > | 25% perturbation | $d_i/d_a$ |  20.5731  | 1.0082  | 6.9559 | 29.4320  | 19.3459  | 7.7016  | 16.7429  | 23.0523 | 14.1050 | 0.4837  |
> > > > > |          | $d_i/d_n$ |  4.2974  | 9.3829  | 37.7505 | 68.5836 |12.8552 | 5.1918 | 8.7394  | 7.9608  | 7.9395 |  31.4415  |
> > > > >
> > > > > As we can see, there is no consistent pattern in the numerical data presented in this table. Nevertheless, RQGNN successfully learns the Rayleigh Quotient of the graphs and effectively distinguishes between the two classes. This demonstrates that a consistent pattern in the ratio is not necessary for accurate classification. The key to differentiating between the classes is capturing and leveraging such gaps on certain dimensions, rather than relying on consistent patterns in the data.
> > > > >
> > > > > Furthermore, as we introduce more perturbations to the graphs, the difference in the normalized Rayleigh Quotient of different classes becomes larger. Notice that when we inject 15% perturbation to the graphs, which is already a significant update to the graph, we observe 1-2 times changes in the normalized Rayleigh Quotient values compared to the original values on certain dimensions. These findings are consistent with our reported results in Figure 1 and Figures 7-15, indicating that the observed deviation of the normalized Rayleigh Quotient is significant enough to be captured by our model, thereby improving the task effectiveness.
> > > > >
> > > > > We sincerely hope that we have addressed your concerns satisfactorily, and we will gladly answer any additional questions you may have.
> > > > >
> > > > > ---

---

> > > > > > ### Comment · Reviewer_X1rz · 2023-11-22
> > > > > >
> > > > > > Thank you to the author for addressing my concerns. I recommend that the author consider revising Figure 1, as well as Figures 7-15, as they appear to be confusing. Additionally, I suggest incorporating the rebuttal stage experiments into the manuscript, as this would significantly enhance the persuasiveness of the article. I acknowledge the author's efforts in providing rebuttals and presenting numerous experiments to convince me, and I am open to raising my rating to 6.

---

> > > > > > > ### Author Response · Authors · 2023-11-22
> > > > > > > **We would like to express our gratitude for your valuable comments**
> > > > > > >
> > > > > > > Thanks for your constructive suggestions. We will revise Figure 1, as well as Figures 7-15. Besides, we will incorporate the experiments and discussions above into our revised manuscript.

---

### Official Review · Reviewer_oJCz · 2023-10-31

**Soundness:** 3 good
**Presentation:** 3 good
**Contribution:** 3 good
**Rating:** 6
**Confidence:** 4

**Summary:**

In this paper, the authors introduce spectral analysis and identify significant differences in the spectral energy distributions between anomalous and normal graphs, leading to the development of the Rayleigh Quotient Graph Neural Network (RQGNN). This approach combines the explicit capture of the Rayleigh Quotient and implicit spectral exploration, outperforming existing methods in comprehensive experiments.

**Strengths:**

1. This paper is well-organized and easy to follow.
2. Investigate Rayleigh Quotient Learning to graph anomaly detection is promising.
3. The experiment demonstrates the effectiveness of the proposed method.

**Weaknesses:**

1. The reviewer thinks that the motivation and rationale of Rayleigh Quotient Learning should emphasized. It is a quite general problem that “existing methods fail to capture the underlying properties of graph anomalies”.
2. An algorithm describing the training process is required.
3. From the objective function in Section 3, the method proposed in this paper is a supervised method. This implies that some of the comparisons in the experiments are unfair because many graph-level anomaly detection methods are fully unsupervised to my knowledge, e.g., OCGIN, OCGTL, GlocalKD, etc.
4. The authors only test on chemical datasets in this paper, while there are other data types that are constructed as graphs, such as social networks.
5. It seems that the proposed RQGNN fluctuates a lot with the change of hyperparameters and shows instability. The authors should explain the reasons for this observation.
6. The impact of hyperparameters on loss function is encouraged to be explored.
7. From the ablation study results, the performance of RQGNN does not consistently outperform the other degradation models, and their performance is quite close in many cases. It seems the improvement from the several proposed components is somewhat limited.

**Questions:**

1. Can the Rayleigh Quotient Learning be used to detect node anomalies?
2. The authors should show the ratio between normal and anomalous graphs. Are they balanced or not?

---

> ### Author Response · Authors · 2023-11-19
> **Author Response (1/2)**
>
> We appreciate your comprehensive and constructive review. Your crucial comments on experiments are exceedingly helpful for us to improve our manuscript. Our point-to-point responses to your comments are given below.
>
> ---
>
> **W1**: The reviewer thinks that the motivation and rationale of Rayleigh Quotient Learning should be emphasized. It is a quite general problem that “existing methods fail to capture the underlying properties of graph anomalies”.
>
> **RW1**: Thanks for your constructive suggestion. We will revise the claim as “existing methods fail to capture the property of Rayleigh Quotient in graph anomalies” to emphasize it.
>
> As you can observe from Figure 1 in Section 1, a noticeable distinction between the Rayleigh Quotient of the normal graphs and that of the anomalous graphs can be observed. Subsequent experiments have demonstrated that using this characteristic significantly boosts the performance of graph-level anomaly detection tasks. To avoid confusion, we have revised the manuscript as “the spectral property of graph anomalies”. Furthermore, we will incorporate an additional table to emphasize the importance of the Rayleigh Quotient in graph-level anomaly detection. For more detailed information, please refer to **RW1 (Reviewer X1rz)**.
>
> ---
>
> **W2**: An algorithm describing the training process is required.
>
> **RW2**: Thanks for your valuable suggestion. We will include an algorithm in the Appendix of our revised manuscript to provide a detailed description of the training process.
>
> ---
>
> **W3**: From the objective function in Section 3, the method proposed in this paper is a supervised method. This implies that some of the comparisons in the experiments are unfair because many graph-level anomaly detection methods are fully unsupervised to my knowledge, e.g., OCGIN, OCGTL, GlocalKD, etc.
>
> **RW3**: Yes, almost all graph-level anomaly detection methods tested in our paper are fully unsupervised. Indeed, the lack of fairness in this area is primarily due to its under-exploration. The main reason is the scarcity of supervised learning methods that can serve as baseline models. To the best of our knowledge, iGAD is the only supervised learning model available for the graph-level anomaly detection task. Consequently, we follow the experimental settings of iGAD. Furthermore, we include more baselines that are related to this task to demonstrate the effectiveness of our proposed model. The existence of this phenomenon also highlights the urgent need to develop new and robust supervised learning models, showing the importance of our work.
>
> ---
>
> **W4**: The authors only test on chemical datasets in this paper, while there are other data types that are constructed as graphs, such as social networks.
>
> **RW4**: Since there is currently no existing dataset specifically designed for graph-level anomaly detection in other domains, we modify a synthetic dataset (COLORS-3) and a social network classification dataset (DBLP_v1) to conduct additional experiments. For more detailed information regarding this set of experiments, please refer to **RW2 (Reviewer qWnD)**.
>
> ---
>
> **W5**: It seems that the proposed RQGNN fluctuates a lot with the change of hyperparameters and shows instability. The authors should explain the reasons for this observation.
>
> **RW5**: Despite observing variations in the model performance, our model still outperforms other baselines in almost all datasets. Notice that other baselines, such as iGAD, also display performance variations in their experiments when varying hyperparameters.
>
> ---

---

> ### Author Response · Authors · 2023-11-19
> **Author Response (2/2)**
>
> **W6**: The impact of hyperparameters on loss function is encouraged to be explored.
>
> **RW6**: Thanks for your valuable suggestion. Due to the time limit, we conduct experiments of parameter analysis on four datasets. The experimental results of varying hyperparameters are as follows. To summarize, when the $\gamma$ is set to 1.5, RQGNN achieves a relatively stable performance on these datasets. Meanwhile, RQGNN presents a relatively satisfactory when $\beta$ is set to 0.999.
>
> The results of varying the hyperparameter $\beta$:
>
> | Datasets | Metrics |  0.99  | 0.999  | 0.9999 | 0.99999 |
> | :------: | :-----: | :----: | :----: | :----: | :-----: |
> |  MCF-7   |   AUC   | 0.8454 | 0.8354 | 0.8213 | 0.8381  |
> |          |   F1    | 0.7226 | 0.7394 | 0.7281 | 0.7050  |
> |  SF-295  |   AUC   | 0.8680 | 0.8825 | 0.8674 | 0.8701  |
> |          |   F1    | 0.7159 | 0.7416 | 0.7509 | 0.7142  |
> |  SN12C   |   AUC   | 0.8889 | 0.8861 | 0.8845 | 0.8874  |
> |          |   F1    | 0.7522 | 0.7660 | 0.7466 | 0.7294  |
> | UACC257  |   AUC   | 0.8544 | 0.8724 | 0.8630 | 0.8621  |
> |          |   F1    | 0.6928 | 0.7362 | 0.7323 | 0.6712  |
>
> The results of varying the hyperparameter $\gamma$:
>
> | Datasets | Metrics |  1.0   |  1.5   |  2.0   |  2.5   |
> | :------: | :-----: | :----: | :----: | :----: | :----: |
> |  MCF-7   |   AUC   | 0.8297 | 0.8354 | 0.8570 | 0.8445 |
> |          |   F1    | 0.7433 | 0.7394 | 0.7340 | 0.7187 |
> |  SF-295  |   AUC   | 0.8800 | 0.8825 | 0.8773 | 0.8838 |
> |          |   F1    | 0.7499 | 0.7416 | 0.7329 | 0.7473 |
> |  SN12C   |   AUC   | 0.8817 | 0.8861 | 0.8774 | 0.8851 |
> |          |   F1    | 0.7593 | 0.7660 | 0.7603 | 0.7319 |
> | UACC257  |   AUC   | 0.8710 | 0.8724 | 0.8616 | 0.8599 |
> |          |   F1    | 0.7117 | 0.7362 | 0.7107 | 0.7107 |
>
> We will add these experimental results in the Appendix of our revised manuscript.
>
> ---
>
> **W7**: From the ablation study results, the performance of RQGNN does not consistently outperform the other degradation models, and their performance is quite close in many cases. It seems the improvement from the several proposed components is somewhat limited.
>
> **RW7**: Not exactly. In contrast, the conclusion can be quite the opposite. Our experiments reveal that both RQGNN-1 and RQGNN-2 achieve improvements compared to other baselines, clearly demonstrating the effectiveness of each proposed component in boosting graph-level anomaly detection performance. By combining these two powerful components, we get our proposed RQGNN model.
>
> As we can observe from experiments conducted in Section 4, there are several reasons to highlight the necessity of combining these two components:
> RQGNN vs. RQGNN-1
> RQGNN outperforms RQGNN-1 in all the datasets in terms of both metrics.
> RQGNN vs. RQGNN-2
> RQGNN outperforms RQGNN-2 in 7 datasets in terms of both metrics.
> Although RQGNN-2 slightly outperforms RQGNN in 3 datasets in terms of one metric, it significantly falls behind RQGNN in terms of the other metric.
>
> These findings clearly demonstrate the necessity of combining both components to achieve superior and more stable performance in the graph-level anomaly detection task. Besides, we aim to find the unified hyperparameters that can achieve good performance for all datasets.
>
> In summary, the proposed two components demonstrate their individual strengths, and we can even combine them together to achieve superior and more stable performance. It is evident that the improvement is not limited, instead, each of the proposed components will provide a new design direction for future research in this area.
>
> ---
>
> **Q1**: Can the Rayleigh Quotient Learning be used to detect node anomalies?
>
> **RQ1**: Yes, the Rayleigh Quotient can be used to detect node anomalies. In the original manuscript, we have cited Tang et al. [7] where the authors introduce an interesting idea for node anomaly detection related to the Rayleigh Quotient. However, their focus primarily lies on designing a spectral GNN for node anomaly detection, rather than delving into the properties of the Rayleigh Quotient in the context of anomaly detection. As shown in our paper, Rayleigh Quotient can reveal the properties underlying anomalies and can be further explored in the GNN design for node-level anomaly detection.
>
> [7] Jianheng Tang, Jiajin Li, Ziqi Gao, and Jia Li. Rethinking graph neural networks for anomaly detection. In ICML, pp. 21076–21089, 2022.
>
> ---
>
> **Q2**: The authors should show the ratio between normal and anomalous graphs. Are they balanced or not?
>
> **RQ2**: We have provided the number of normal and anomalous graphs of each dataset in Table 1 in the original manuscript. The anomalous ratio of each dataset is around 5%, which shows that these datasets are highly imbalanced. Please refer to Table 1 in Section 4 for more details.
>
> ---
>
> We sincerely appreciate your time, and we are glad to answer any additional questions you may have.

---

> > ### Comment · Reviewer_qWnD · 2023-11-19
> >
> > Thank you for the clarifications and the additional numerical experiments.
> >
> > While "out of distribution" seems easy for common understanding, defining an anomaly might easily differ from author to author.  Perhaps include a sentence or two as in your response to help the reader.
> >
> > I still believe Q5 above is an important theoretical question that also gets to generality.  It seems the key question is how much perturbation is detectable, starting with some given graph class.  And has this problem been studied, given the large literature in physics, statistics, and signal processing.

---

> > > ### Author Response · Authors · 2023-11-21
> > > **Author Response (1/1)**
> > >
> > > ---
> > >
> > > **C1**: While "out of distribution" seems easy for common understanding, defining an anomaly might easily differ from author to author. Perhaps include a sentence or two as in your response to help the reader.
> > >
> > > **RC1**: Thanks for your constructive suggestion. We will include a description of the difference between anomaly detection and out-of-distribution detection in our revised paper to help the readers understand.
> > >
> > > ---
> > >
> > > **C2**:  I still believe Q5 above is an important theoretical question that also gets to generality. It seems the key question is how much perturbation is detectable, starting with some given graph class. And has this problem been studied, given the large literature in physics, statistics, and signal processing.
> > >
> > > **RC2**: Thanks for your comments. To the best of our knowledge, no previous work has applied the Rayleigh Quotient to graph-level anomaly detection. Although there are some related studies about perturbation and Rayleigh Quotient in theory, the best result is to provide a boundary of change of the Rayleigh Quotient given a small perturbation, as shown in Section 3 of our original manuscript. However, how much perturbation is detectable depends on the models we use to detect the anomalies. Due to the black-box nature of deep learning models, it is hard to provide a theoretical analysis in terms of how much perturbation is detectable. Nevertheless, in practice, our RQGNN manages to learn the differences between the two classes and achieves superior results compared with other baselines. This shows that the graph-level anomaly detection model will benefit from exploring the properties of Rayleigh Quotient. Besides, as the case study shown in Section 4, even a slight change in the Rayleigh Quotient can be detected by RQGNN, which further proves the effectiveness of the Rayleigh Quotient on graph-level anomaly detection.
> > >
> > > To explore how much perturbation is detectable by RQGNN, we further conduct experiments on SN12C. We use SN12C to create 5 new synthetic datasets. Specifically, we first randomly select 5% of normal graphs to perform perturbations. For these graphs, each edge is changed with probability $p$, where $p=0.05, 0.10, 0.15, 0.20,$ or $0.25$. Then we use these perturbed samples as anomalous graphs and all the remaining 95% normal graphs in SN12C to construct these five new datasets. We present the results in the following table:
> > >
> > > | Metrics |   0.05  |  0.10 | 0.15 |  0.20  | 0.25 |
> > > | :-----: | :--------: | :----: | :----: | :----: | :----: |
> > > |   AUC   | 0.6525 | 0.7592 | 0.9736 | 0.9884 |  0.9798   |
> > > |   F1    | 0.6966 | 0.8126 | 0.9113 | 0.9471 | 0.9623 |
> > >
> > > As we can see from the table, Even if the perturbation probability is only 0.05, RQGNN can still achieve a reasonable result. When the probability of perturbation reaches 0.15, our RQGNN achieves AUC around 1 and F1 over 0.9, which means RQGNN can nicely distinguish the two classes.
> > >
> > > ---
> > >
> > > We sincerely hope that we have addressed your concerns satisfactorily, and we will gladly answer any additional questions you may have.

---

> ### Author Response · Authors · 2023-11-22
> **We would like to express our gratitude for your valuable comments**
>
> We sincerely hope that we have addressed your concerns satisfactorily, and we will gladly answer any additional questions you may have.

---

### Official Review · Reviewer_qWnD · 2023-11-01

**Soundness:** 2 fair
**Presentation:** 3 good
**Contribution:** 2 fair
**Rating:** 6
**Confidence:** 4

**Summary:**

This paper considers the problem of detecting anomalies in graphs.  Related work in GNNs is reviewed, and it is argued that the Rayleigh quotient (RQ) supplies feature information that is useful and hasn’t been adequately explored.  This is based on some data exploration of a chemical dataset.  The authors set up a learning problem that incorporates the RQ, and incorporates several pieces including graph wavelets and using the RQ as an attention mechanism.  Experiments compare baseline GNNs, including GNN classifiers and GNN anomaly detectors, and show the benefits of the proposed approach for the datasets studied.

**Strengths:**

The paper is well written and provides a good statement of the problem, prior GNN related work, and the motivation for the approach.   Exploring the Rayleigh quotient in this context is well motivated and interesting.

The paper has some nice innovation, incorporating RQ pooling for example, and the use of RQ as a means of attention in the context of the graph wavelet.  The use of wavelets seems well motivated for this “step change” problem, when the change is an anomaly.

The comparisons with other GNN-based methods are well documented and show the benefit of the proposed method for the datasets considered.  The hyperparameter choices are well described, and the ablation studies are useful indicators.

**Weaknesses:**

The relation between Rayleigh quotient and perturbation is well known and studied, for example, in physics.  For example:  Pierre, C. (December 1, 1988). "Comments on Rayleigh’s Quotient and Perturbation Theory for the Eigenvalue Problem." ASME. J. Appl. Mech. December 1988; 55(4): 986–988. When the vector is close to an eigenvector, then the RQ has a value that is close to the corresponding eigenvector.  There is also the Rayleigh-Schrödinger procedure that yields approximations to the eigenvalues and eigenvectors of a perturbed matrix by a sequence of successively higher order corrections to the eigenvalues and eigenvectors of the unperturbed matrix.

The paper explores an important application and datasets, and experimentally shows that the RQ provides a useful feature for detecting change.  However, it isn’t clear how general this is beyond the application considered.

The anomaly here is a change detector between the baseline distribution (the normal graph class), and some deviation from this graph.  Apparently for this data there is no clear class after change (so it isn’t surprising that the graph classifiers don’t work well).

**Questions:**

Given the large literature on RQ’s it seems likely that eqn (1) is well known?

Lemma 1 and leading to eqn (4), using L – I_n to compute is from Rivlin or some other reference?

Section 4.5 (and throughout the paper).  “Rayleigh Quotient is an intrinsic characteristic of the graph-level anomaly detection task.”  Should this be altered to say “for the application studied”?  How general is the claim?

Isn’t the anomaly for this application the same as an “out of distribution” test?

It seems this area of study would benefit from good baseline data sets.  For example, for what random classes of graphs is the RQ approach well founded?  This could be studied through simulation and theory.

Section 4.1: Perhaps you could say a little more about “various chemical compounds and their reactions to different cancer cells”, and how a chemical leads to a graph?

---

> ### Author Response · Authors · 2023-11-19
> **Author Response (1/2)**
>
> We appreciate your comprehensive and constructive review. Your crucial comments on experiments are helpful for us to improve our manuscript. Our point-to-point responses to your comments are given below.
>
> ---
>
> **W1**: The relation between Rayleigh quotient and perturbation is well known and studied, for example, in physics. For example: Pierre, C. (December 1, 1988). "Comments on Rayleigh’s Quotient and Perturbation Theory for the Eigenvalue Problem." ASME. J. Appl. Mech. December 1988; 55(4): 986–988. When the vector is close to an eigenvector, then the RQ has a value that is close to the corresponding eigenvector. There is also the Rayleigh-Schrödinger procedure that yields approximations to the eigenvalues and eigenvectors of a perturbed matrix by a sequence of successively higher order corrections to the eigenvalues and eigenvectors of the unperturbed matrix.
>
> **RW1**: Thanks for bringing that up. We acknowledge that the relation between Rayleigh Quotient and perturbation is well known and studied in physics. However, the inherent properties of the Rayleigh Quotient in the graph domain are still relatively under-explored. Furthermore, to the best of our knowledge, no previous work has considered using Rayleigh Quotient for the graph-level anomaly detection task. Our key observation of Rayleigh Quotient on graph-level anomaly detection as shown in Figure 1 in the original manuscript can provide a new direction for this under-explored field. We will ensure to cite these related papers in our revised manuscript.
>
> ---
>
> **W2**: The paper explores an important application and datasets, and experimentally shows that the RQ provides a useful feature for detecting change. However, it isn’t clear how general this is beyond the application considered. The anomaly here is a change detector between the baseline distribution (the normal graph class), and some deviation from this graph. Apparently for this data there is no clear class after change (so it isn’t surprising that the graph classifiers don’t work well).
>
> **RW2**: Thanks for the constructive suggestions. Indeed, although the main application of graph-level anomaly detection is typically focused on detecting chemical compounds, there are still applications in other domains such as social networks. However, due to the lack of datasets specifically collected for graph-level anomaly detection in other domains, we construct our own datasets and conduct experiments on them. To be comprehensive, we use two graph classification datasets to evaluate our RQGNN, one is COLORS-3 [1], which is a synthetic dataset in the TUDataset repository [2], and the other is DBLP_v1 [3], which is a social network dataset as classified in TUDataset [2].
>
> Specifically, we first conduct experiments on the COLORS-3 [1] dataset, which contains 11 classes. We use 10 classes to form normal graphs and use the remaining class as anomalous graphs. The experimental results are as follows:
>
> | Metrics |    RQGNN    |  iGAD  | HimNet |  GMT   | TVGNN  | BernNet | ChebyNet |
> | :-----: | :--------: | :----: | :----: | :----: | :----: | :-----: | :------: |
> |   AUC   | **0.9378** | 0.7385 | 0.5129 | 0.5063 |  0.5000   | 0.6136  |  0.6192  |
> |   F1    | **0.7640** | 0.5301 | 0.5934 | 0.4764 | 0.4764 | 0.4764  |  0.4764  |
>
> Then, we conduct additional experiments in the social network domain using the DBLP_v1 [3] dataset, which is a well-balanced social network classification dataset in the field of computer science. In this dataset, each graph denotes a research paper belonging to either DBDM (database and data mining) or CVPR (computer vision and pattern recognition) field, where each node denotes a paper ID or a keyword and each edge denotes the citation relationship between papers or keyword relations in the title. To create a new graph-level anomaly detection dataset, we randomly sample 5% of one class as the anomaly class and use the other class as the normal class. The experimental results are as follows:
>
> | Metrics |    RQGNN    |  iGAD  | HimNet |  GMT   | TVGNN  | BernNet | ChebyNet |
> | :-----: | :--------: | :----: | :----: | :----: | :----: | :-----: | :------: |
> |   AUC   | **0.8808** | 0.7346 | 0.6656 | 0.8234 | 0.8455 | 0.8549  |  0.8369  |
> |   F1    | **0.7709** | 0.6648 | 0.6133 | 0.4877 | 0.6449 | 0.4877  |  0.5472  |
>
> Compared with other baseline models, our proposed RQGNN achieves significant improvements in both AUC and F1 scores in these two new domains, which demonstrates the effectiveness of our RQGNN in detecting anomalies beyond the application considered.
>
> [1] Boris Knyazev, Graham W. Taylor, Mohamed R. Amer. Understanding attention and generalization in graph neural networks. NeurIPS, pages 4204-4214, 2019.
>
> [2] https://chrsmrrs.github.io/datasets/docs/datasets/
>
> [3] Shirui Pan, Xingquan Zhu, Chengqi Zhang, and Philip S. Yu. Graph Stream Classification using Labeled and Unlabeled Graphs, ICDE, pages 398-409, 2013.
>
> ---

---

> ### Author Response · Authors · 2023-11-19
> **Author Response (2/2)**
>
> **Q1**: Given the large literature on RQ’s it seems likely that eqn (1) is well known?
>
> **RQ1**: Yes. We acknowledge that the relation between Rayleigh Quotient and perturbation is well known and studied in physics. However, the inherent properties of the Rayleigh Quotient in the graph domain are still relatively under-explored. Furthermore, to the best of our knowledge, no previous work has considered using Rayleigh Quotient for the graph-level anomaly detection task. Our key observation of Rayleigh Quotient on graph-level anomaly detection as shown in Figure 1 in the original manuscript can provide a new direction for this under-explored field.
>
> ---
>
> **Q2**: Lemma 1 and leading to eqn (4), using L – I_n to compute is from Rivlin or some other reference?
>
> **RQ2**: The theory of Chebyshev’s series comes from Rivlin et al. [4], and the Eqn. 4 is mainly from Hammond et al. [5]. We will cite this paper in our revised manuscript.
>
> [4] GTheodore J. Rivlin. The Chebyshev polynomials. John Wiley & Sons, 1974.
>
> [5] Hammond D K, Vandergheynst P, Gribonval R. Wavelets on graphs via spectral graph theory[J]. Applied and Computational Harmonic Analysis, 2011, 30(2): 129-150.
>
> ---
>
> **Q3**: Section 4.5 (and throughout the paper). “Rayleigh Quotient is an intrinsic characteristic of the graph-level anomaly detection task.” Should this be altered to say “for the application studied”? How general is the claim?
>
> **RQ3**: We admit that we should add the description to weaken the claim and we have revised the description as “Rayleigh Quotient is an intrinsic characteristic of the graph-level anomaly detection task for the application studied”. However, in addition to its application in detecting chemical compounds, Rayleigh Quotient can also be applied in various other domains such as social network areas.  Please refer to **RW2** for more details.
>
> ---
>
> **Q4**: Isn’t the anomaly for this application the same as an “out of distribution” test?
>
> **RQ4**: Not exactly, there are notable distinctions between anomaly detection and out-of-distribution detection.
>
> Typically, in out-of-distribution datasets, we only have one class called in-distribution data, so out-of-distribution detection focuses on determining whether a new sample belongs to the in-distribution data. However, anomaly datasets usually contain data with two distinct characteristics. Therefore, anomaly detection aims to determine that a new sample belongs to one of the two classes.
>
> For example, anomaly detection involves identifying all the tigers from a dataset that contains both cats and tigers, whereas out-of-distribution detection focuses on determining whether a new cat sample, such as a Birman cat, can be correctly classified as a cat given training dataset that exclusively contains Ragdoll cats.
>
> ---
>
> **Q5**: It seems this area of study would benefit from good baseline data sets. For example, for what random classes of graphs is the RQ approach well founded? This could be studied through simulation and theory.
>
> **RQ5**: We conduct additional experiments on both a synthetic dataset and a social network classification dataset. For more detailed information regarding these experiments, please refer to **RW2**.
>
> ---
>
> **Q6**: Section 4.1: Perhaps you could say a little more about “various chemical compounds and their reactions to different cancer cells”, and how a chemical leads to a graph?
>
> **RQ6**: The raw data is collected from the PubChem website [6]. This website provides comprehensive information on the biological activities of small molecules, including bioassay records for anti-cancer screen tests with different cancer cell lines. Each dataset corresponds to a specific type of cancer screen with the outcomes categorized as either active or inactive, where the active instances are labeled as 1 and the inactive ones are labeled as 0. Please refer to the Appendix for detailed information on the specific tumor associated with each dataset.
>
> To represent the chemical compounds as graphs, we can consider the atoms in the chemical compounds as nodes, and the chemical bonds between two atoms as edges. Therefore, each chemical compound can be represented as a graph.
>
> We will include these discussions in the Appendix of our revised manuscript.
>
> [6] http://pubchem.ncbi.nlm.nih.gov.
>
> ---
>
> We sincerely appreciate your time, and we are glad to answer any additional questions you may have.

---

> ### Author Response · Authors · 2023-11-22
> **We would like to express our gratitude for your valuable comments**
>
> We sincerely hope that we have addressed your concerns satisfactorily, and we will gladly answer any additional questions you may have.

---

> ### Author Response · Authors · 2023-11-23
> **Update for the response of Q5**
>
> ---
>
> **Q5**: It seems this area of study would benefit from good baseline data sets. For example, for what random classes of graphs is the RQ approach well founded? This could be studied through simulation and theory.
>
> **RQ5**: To further address your concern, we show how much perturbation is detectable by RQGNN by simulation. We use SN12C to create 5 new synthetic datasets. Specifically, we first randomly select 5% of normal graphs to perform perturbations. For these graphs, each edge is changed with probability $p$, where $p=0.05, 0.10, 0.15, 0.20,$ or $0.25$. Then we use these perturbed samples as anomalous graphs and all the remaining 95% normal graphs in SN12C to construct these five new datasets. We present the results in the following table:
>
> | Metrics |   0.05  |  0.10 | 0.15 |  0.20  | 0.25 |
> | :-----: | :--------: | :----: | :----: | :----: | :----: |
> |   AUC   | 0.6525 | 0.7592 | 0.9736 | 0.9884 |  0.9798   |
> |   F1    | 0.6966 | 0.8126 | 0.9113 | 0.9471 | 0.9623 |
>
> As we can see from the table, Even if the perturbation probability is only 0.05, RQGNN can still achieve a reasonable result. When the probability of perturbation reaches 0.15, our RQGNN achieves AUC around 1 and F1 over 0.9, which means RQGNN can nicely distinguish the two classes.
>
> ---
>
> We sincerely hope that we have addressed your concerns satisfactorily, and we will gladly answer any additional questions you may have.

---

### Author Response · Authors · 2023-11-21
**Modifications in the revised version of our manuscript.**

We express our gratitude to all the reviewers for their thorough and constructive feedback. Taking into consideration the valuable comments provided by the reviewers, we intend to incorporate the following modifications in the revised version of our manuscript.

---

Abstract
- revise the claim. (Reviewer **oJCz W1**)

Section 1
- add some discussions about the Rayleigh Quotient. (Reviewer **qWnD W1** and Reviewer **oJCz Q1**)

Section 2
- add brief introduction about the Rayleigh Quotient (Reviewer **X1rz W2**)

- add some discussions about the differences between out-of-distribution and anomaly detection. (Reviewer **oJCz C1**)

Section 3
- add additional citations of using $L-I_n$. (Reviewer **qWnD Q2**)

- add discussions for using graph wavelet convolution. (Reviewer **X1rz W4**)

Section 4
- move the table that lists the statistics of 10 real-world datasets to Appendix A.3
- revise the claim. (Reviewer **qWnD Q3**)

Appendix
- add the experimental results of BernNet-CB and ChebyNet-CB in Table 2 in Appendix A.2. (Reviewer **X1rz Q1**)

- add the description of how to generate graphs from the chemical compounds in Appendix A.3. (Reviewer **qWnD Q6**)

- add additional experiments on datasets beyond the biography domain in Appendix A.4. (Reviewer **qWnD W2** and Reviewer **X1rz W3**)

- add algorithms in Appendix A.5. (Reviewer **oJCz W2**)

- add parameter analysis in Appendix A.6. (Reviewer **oJCz W6**)

- add experiments to explore how much perturbation is detectable by the RQGNN in Appendix A.7. (Reviewer **oJCz C2** and Reviewer **X1rz C2**)

---

These modifications have been included in the revised version of our manuscript, which has been highlighted in blue to facilitate the reviewing process.

---

### Meta-Review · Area_Chair_oUei · 2023-12-10

**Metareview:**

I recommend to accept this paper.

  In this paper, the authors introduced spectral analysis to observed differences in abnormal vs. normal graphs. Rayleigh Quotient Graph Neural Network (RQGNN) excels in trials, merging explicit and implicit spectral insights.

  In this paper, authors addresses graph-level anomaly detection challenges by exploring spectral differences. Based on the observation of disparities in spectral energy between normal and anomalous graphs, the authors introduces RQGNN, the first spectral GNN for graph-level anomaly detection.

  After rebuttal, most of the concerns and questions raised by reviewers were properly addressed. As a result, all the reviewers recommended to accept this paper. I would encourage the authors to take feedback and suggestions from reviewers in the preparation of the camera ready.

**Justification For Why Not Higher Score:**

N/A.

**Justification For Why Not Lower Score:**

It is a good paper. The authors made great effort to resolve the concerns from by two reviewers who gave lower scores at the beginning and convince them to raise the score to 6 after rebuttal.

---

### Decision · Program_Chairs · 2024-01-16

Accept (poster)